# Targeting oncogenic KRasG13C with nucleotide-based covalent inhibitors

Lisa Goebel[1], Tonia Kirschner[1], Sandra Koska[1], Amrita Rai[2], Petra Janning[3], Stefano Maffini[4], Helge Vatheuer[1], Paul Czodrowski[1], Roger S Goody[2]*, Matthias P Müller[1]*, Daniel Rauh[1]*

[1]Department of Chemistry and Chemical Biology, TU Dortmund University, Dortmund, Germany; [2]Department of Structural Biochemistry, Max Planck Institute of Molecular Physiology, Dortmund, Germany; [3]Department of Chemical Biology, Max Planck Institute of Molecular Physiology, Dortmund, Germany; [4]Department of Mechanistic Cell Biology, Max Planck Institute of Molecular Physiology, Dortmund, Germany

**Abstract** Mutations within Ras proteins represent major drivers in human cancer. In this study, we report the structure-based design, synthesis, as well as biochemical and cellular evaluation of nucleotide-based covalent inhibitors for KRasG13C, an important oncogenic mutant of Ras that has not been successfully addressed in the past. Mass spectrometry experiments and kinetic studies reveal promising molecular properties of these covalent inhibitors, and X-ray crystallographic analysis has yielded the first reported crystal structures of KRasG13C covalently locked with these GDP analogues. Importantly, KRasG13C covalently modified with these inhibitors can no longer undergo SOS-catalysed nucleotide exchange. As a final proof-of-concept, we show that in contrast to KRasG13C, the covalently locked protein is unable to induce oncogenic signalling in cells, further highlighting the possibility of using nucleotide-based inhibitors with covalent warheads in KRasG13C-driven cancer.

*For correspondence:
roger.goody@mpi-dortmund.mpg.de (RSG);
matthias3.mueller@tu-dortmund.de (MPM);
daniel.rauh@tu-dortmund.de (DR)

**Competing interest:** The authors declare that no competing interests exist.

## Editor's evaluation

The authors present important information regarding the possibility of targeting the oncogenic K-Ras(G13C) mutant with nucleotide competitors. The experiments represent a solid support of the claims and show that this approach can work despite concerns about the high affinity of GTP and its high cellular concentration. These results will be of high interest for all working in the Ras field and in targeting oncogenes with small molecules. A weakness of the manuscript is the lack of direct physiological insights.

## Introduction

Ras proteins act as key regulators of many cellular processes by switching between inactive GDP-bound and active GTP-bound states, the latter specifically activating several downstream signalling pathways (*Cox et al., 2014*). Oncogenic Ras mutations that lead to dysregulation of the switch mechanism are found in about 25% of all human cancers, including three of the most lethal forms (lung, colon, and pancreatic cancer). Among the Ras proteins, KRas is the predominantly mutated isoform (85%), followed by NRas (11%) and HRas (4%), with mutational hotspots at amino acid positions G12, G13, and Q61 (*Cox et al., 2014*; *Hobbs et al., 2016*). Although the glycine at position 12 is the most commonly mutated residue, G13 is the second most common mutation (14% of tumors harbor a mutation at this position) and in 6% of these cases, an acquired cysteine is found (*Forbes et al.,*

2015; *Visscher et al., 2016*). In lung cancer, the prevalence of the G13C mutation is 3%, which is equivalent to approximately 7000 individuals in the US per year (*Forbes et al., 2015*; *Burge and Hobbs, 2022*). Because of their prominent role in cancer, Ras oncogenes were identified as attractive targets for cancer therapy since their initial discovery in 1981, but attempts to target Ras have been largely unsuccessful and Ras proteins were long considered undruggable. After decades of failure, new interest has recently arisen from selective targeting of the G12C oncogenic mutant of KRas (*Ostrem et al., 2013*; *Patricelli et al., 2016*; *Janes et al., 2018*; *Shin, 2019*; *Canon et al., 2019*; *Hong, 2020*; *Fell et al., 2018*; *Hallin et al., 2020*; *Fell, 2020*). Inhibitors that bind irreversibly to the G12C mutated cysteine residue within a previously unknown switch-II pocket were originally identified and designed in the Shokat laboratory, and have been further developed within the academic and industrial world to advance candidates into the clinic (*clinicaltrials.gov, 2018a*; *clinicaltrials.gov, 2019a*; *clinicaltrials.gov, 2018b*; *clinicaltrials.gov, 2019b*; *Goebel et al., 2020*). In May 2021, the first-in-class KRasG12C inhibitor Sotorasib (Amgen) was approved by the FDA for the treatment of non-small-cell lung cancer (NSCLC), confirming the therapeutic susceptibility of mutant KRas in cancer (*Mullard, 2021*). A second approach for targeting mutant KRas has been described using nucleotide competitive inhibitors that can covalently bind to KRasG12C (*Lim et al., 2014*; *Xiong et al., 2017*). Strategies involving direct competition with nucleotide binding were originally set aside because of the high affinity of GDP/GTP for Ras and high cellular GDP/GTP concentrations. However, the combination of nucleotide competition with covalent binding of the inhibitors to the Ras protein has fuelled new hope. Gray and colleagues developed SML-8-73-1, a GDP derivative harbouring an electrophilic group on the β-phosphate to irreversibly bind to the mutant cysteine at position 12 within the P-loop of Ras (*Lim et al., 2014*). Unfortunately, the modification of the β-phosphate leads to a dramatic loss of affinity because of the loss of important interactions with the protein and the $Mg^{2+}$ ion (*Müller et al., 2017*). In this publication, we demonstrate that GDP/GTP/GppCp analogues with an electrophilic group attached to the ribose interact with the necessary high reversible affinity to KRas and are able to react covalently with KRasG13C.

## Results and discussion

### Selective covalent modification of KRasG13C by 2′,3′-modified nucleotide analogues

Available structural information about the binding of GDP and GTP toward Ras provides a detailed understanding of the underlying high affinity of nucleotides through multiple reversible interactions. Based on published crystal structures of Ras proteins (*Figure 1*) and a multiple sequence alignment of the Ras small GTPase superfamily (*Figure 1—figure supplement 1*, *Supplementary file 1*), we designed and synthesized guanine nucleotide-based inhibitors with an additional Michael acceptor as a covalent warhead for targeting oncogenic KRas variants harboring cysteines in the P-loop (KRasG12C and KRasG13C). Whereas the G12C mutation has been successfully addressed in the past, KRasG13C is a largely unexplored target in cancer therapy. However, based on pKa calculations, we were able to show that the G13C mutation should also be generally addressable by appropriately positioned Michael acceptors (*Supplementary file 2*). In contrast to Gray and colleagues, we chose the 2′,3′-OH groups of the ribose for attachment of the warhead since modifications at this position do not significantly alter nucleotide affinity (*Figure 1A, B*; *Eberth et al., 2005*). Nucleotide derivatives with different linkers (eda: ethylenediamine, pda: propylenediamine, bda: butylenediamine) were synthesised based on published procedures (Method section), including GDP/GTP/GppCp analogues (*R*=OH), resulting in the formation of mixed 2′ and 3′-isomers, as well as dGTP analogues (*R*=H) (*Eberth et al., 2005*; *Cremo et al., 1990*). In addition to acrylamide-bearing nucleotides that could potentially bind irreversibly to cysteine containing P-loop mutants via Michael addition, we prepared acetamide derivatives as non-reactive control analogues (*Figure 1B*). The cysteine light-version of KRas constructs lacking other cysteines (C51S, C80L, C118S) were used for initial MS experiments, which indicated that the acrylamide nucleotide derivatives can selectively react with KRasG13C[1-169] (Cys-light), but not with KRasG12C[1-169] (Cys-light) (*Figure 1C, D, E*). The eda linker led to the most efficient covalent protein modification (*Figure 1C*). This tendency is presumably because of a favourable orientation of the reactive warhead and/or a reduced flexibility. Using dGTP analogues, the rate of covalent protein modification was further increased in the case of the eda derivative, indicating

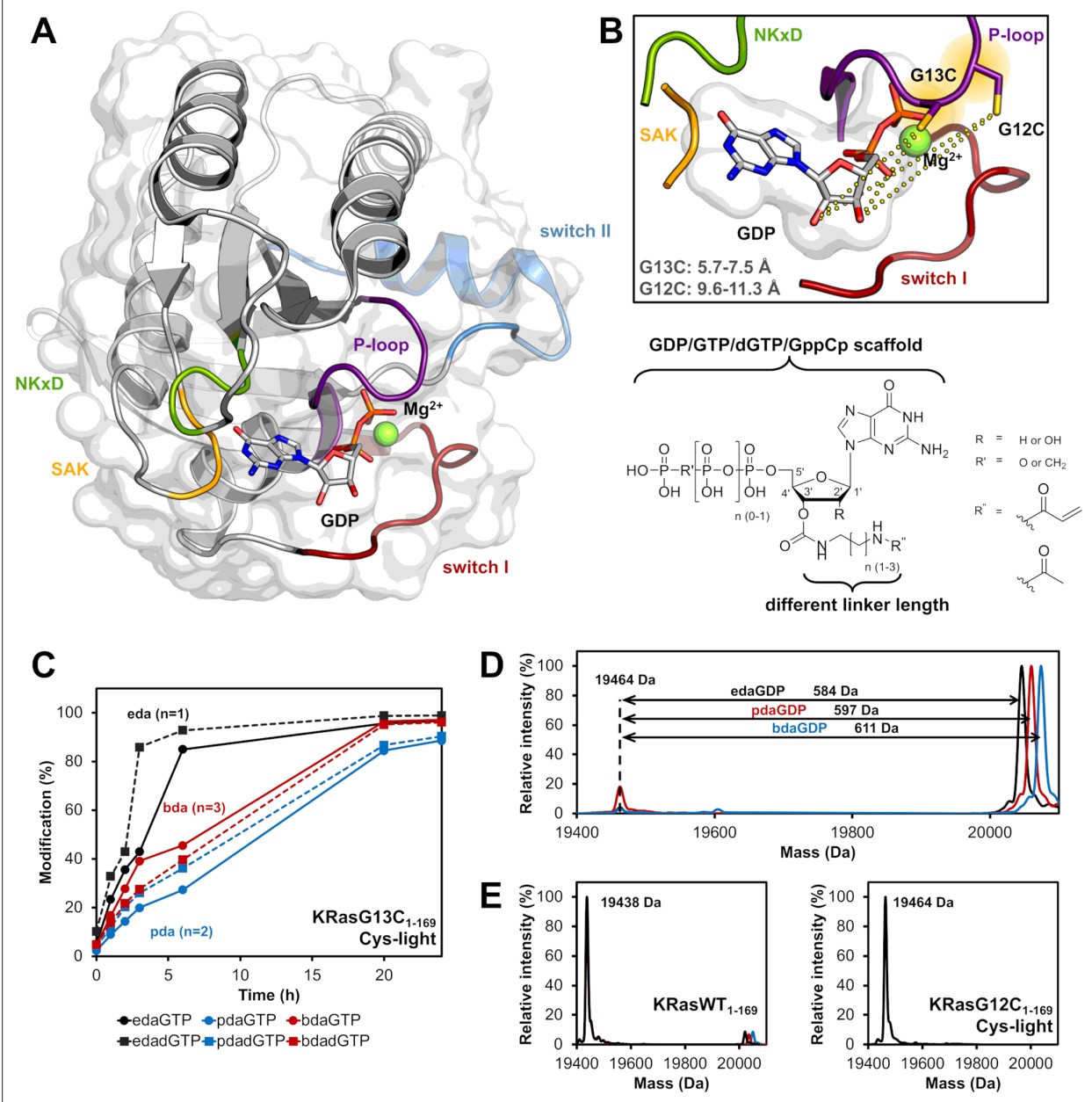

**Figure 1.** Rational design of nucleotide-based covalent KRasG13C inhibitors. (**A**) Structure of KRasWT in the GDP-bound state (grey, P-loop: violet, switch I: red, switch II: blue, NKxD: green, SAK: yellow; PDB 4obe). (**B**) Model of KRasG13C (model is based on PDB 4obe) showing the distances between the cysteine residue and the OH-groups of the ribose and the general structure of nucleotide derivatives bearing an electrophilic group at the 2',3'-position of the ribose moiety, showing that position 12 is further remote compared to position 13 and thus explaining the observed specificity of covalent bond formation. (**C**) Time-dependent analysis of the covalent modification of KRasG13C$_{1-169}$ (Cys-light) at pH 9.5 using different linker lengths (eda: ethylenediamine, pda: propylenediamine and bda: butylenediamine). (**D**) Covalent modification of KRasG13C$_{1-169}$ (Cys-light) proteins at pH 9.5 after 24 hr at room temperature. (**E**) In contrast, KRasG12C$_{1-169}$ (Cys-light) mutant was not modified by nucleotide derivatives and for KRasWT$_{1-169}$ very little unspecific labelling at pH 9.5 was observed compared to the G13C mutant.

The online version of this article includes the following figure supplement(s) for figure 1:

**Figure supplement 1.** Rational design targeting KRasG13C.

**Figure supplement 2.** Time-dependent analysis of the covalent modification of KRasG13C.

**Figure supplement 3.** Time-dependent analysis of the covalent protein modification with edaGDP or edaGTP.

**Figure supplement 4.** Covalent protein modification with edaGDP in the presence of competing GDP/GTP.

**Figure supplement 5.** Covalent protein modification with edaGppCp.

*Figure 1 continued on next page*

*Figure 1 continued*

**Figure supplement 6.** pH-dependent analysis of the covalent protein modification of KRasG13C and KRasWT.

**Figure supplement 7.** Rab21 naturally contains a cysteine at the position equivalent to KRasG13C and becomes modified by edaGDP to an extent comparable to KRasG13C.

that the linker in the 3′-position of the ribose is superior to the 2′-position with respect to targeting KRasG13C. Although, complete modification of KRasG13C was only observed at elevated pH, significant modification of the protein also occurred at a physiological pH within 24 hr. On incubating KRasG13C$_{1-169}$ (Cys-light) with the GTP analogues, we observed at intermediate stages a mixture of covalently bound GDP and GTP forms of KRasG13C$_{1-169}$ (Cys-light), but ultimately the reaction yielded only the diphosphate form, indicating that the nucleotides were still hydrolysed after the covalent reaction and were properly positioned in the active site of KRas (*Figure 1—figure supplement 2*). The time-resolved labelling of KRasG13C$_{1-169}$ (Cys-light) with either GDP or GTP derivatives led to comparable covalent protein modification rates (*Figure 1—figure supplement 3*). Additionally, we tested whether the nucleotides were generally able to compete with their natural counterparts and monitored the reaction also in the presence of equimolar concentrations as well as 10 x and 100 x excess of GDP/GTP (*Figure 1—figure supplement 4*). This shows that although the reaction is slowed down due to the competing nature of GDP and GTP, the modified nucleotides are able to compete. Upon incubating KRasG13C$_{1-169}$ (Cys-light) with a non-cleavable GppCp derivative, we also observed covalent modification, but without subsequent hydrolysis of the nucleotide (*Figure 1—figure supplement 5*). To further investigate the specificity of the reaction towards KRasG13C$_{1-169}$ (Cys-light), we also tested the wild type protein. For KRasWT$_{1-169}$, very little unspecific labelling was observed at pH 9.5 compared to the G13C mutant (approximately 3% at pH 9, 8% at pH 9.5, whereas under equivalent conditions, KRasG13C modification was ≥90%; *Figure 1E*, *Figure 1—figure supplement 6*), thus showing that the warhead reacts preferentially with the cysteine at position 13, but not other cysteines in KRas nor the additional neighboring cysteine in KRasG12C. In addition, the multiple sequence alignment of the Ras small GTPase superfamily revealed that only about 7% of the GTPase members contain cysteines within the P-loop that might potentially be accessible by our linker design and only 3 contain Cys at the position equivalent to residue 13 in Ras (Arl4a, RheBL1, Rab21). Upon incubating Rab21 with edaGDP, we indeed observed similar modification compared to KRasG13C (*Figure 1—figure supplement 7*). However, since cross-reactivities of covalently binding molecules is well-known and documented also for example in kinase inhibitors such as osimertinib, which modifies a number of off-target kinases (*Finlay et al., 2014*), we are confident that this will not generally preclude the usage

**Table 1.** Overview of the calculated kinetic parameters ($K_D$, $k_{on}$ and $k_{off}$).
The $k_{on}$ values were determined from stopped-flow experiments (*Table 1—source data 1* and *Table 1—source data 2*), $K_D$ values of pdaGDP and bdaGDP obtained from an HPLC-based approach (*Table 1—source data 3*) and show the average from the three different experiments in the absence or the presence of EDTA or SOS, respectively (see also *Table 1—source data 4* for the individual results). For reference, the $K_D$ values of GDP and the nucleotide analogue SML-8-73-1 are also listed (*Müller et al., 2017*; *Jeganathan et al., 2018*).

| | $K_D$ [pM] | $k_{on}$ [$\mu M^{-1}s^{-1}$] | $k_{off}$ [$s^{-1}$] |
|---|---|---|---|
| GDP | 2.5 | 4.22 | $1.1 \times 10^{-5}$ |
| pdaGDP | 8.6±1.3 | 3.34 | $2.9 \times 10^{-5}$ |
| bdaGDP | 9.6±0.5 | 3.12 | $3.0 \times 10^{-5}$ |
| SML-8-73-1 | ~140 nM | - | - |

The online version of this article includes the following source data for table 1:

**Source data 1.** Kinetics of the nucleotide association ($k_{on}$).

**Source data 2.** $k_{on}$ calculation.

**Source data 3.** HPLC-based approach for determination of affinities relative to GDP.

**Source data 4.** $K_D$ calculations.

of the nucleotides described. In fact, Rab21 might be an interesting target itself since knock-down of Rab21 has been reported to have beneficial effects in human glioma cells (*Ge et al., 2017*).

In summary, since other cysteines that are located within the P-loop or other regions close to the nucleotide are mostly further remote compared to Cys13 and even the directly neighboring residue at position 12 does not become modified as shown above, suitable design and optimization of the linker will likely allow sufficient specificity towards KRasG13C also *in vivo* (*Figure 1—figure supplement 1*, *Supplementary file 1*).

## Reversible affinities of the nucleotide analogues are comparable to those of the unmodified nucleotides

To evaluate the impact of the attached linker on nucleotide binding, we first determined the affinity and kinetics of the interaction of the nucleotide derivatives with Ras compared to unmodified GDP/GTP that have dissociation constants ($K_D$) in the picomolar range (*Table 1*). For this purpose, we measured the kinetics of the nucleotide association ($k_{on}$) in a stopped-flow instrument using competition experiments between mantdGDP (2 µM) and increasing amounts of competing nucleotides (1, 2, and 6 µM; *Müller et al., 2017*). As shown in *Table 1—source data 1*, the competitive binding of the competing nucleotide and mantdGDP led to a significant decrease in the fluorescence signal because of smaller amounts of mantdGDP binding to KRas. By fitting the data to a previously described model (*Müller et al., 2017*), we obtained the corresponding $k_{on}$ values, and those for the nucleotide analogues were comparable to those of the unmodified nucleotide (*Table 1—source data 2*). To further analyse the ability of the modified nucleotides to compete with GDP/GTP, KRasWT:GDP was mixed with equal amounts of the acetamide derivatives and incubated either for 7 days at room temperature in the absence of EDTA, for 24 hr at 4 °C in the presence of EDTA, or for 1 hr at room temperature in the presence of SOS (guanine nucleotide exchange factor) to increase the rate of nucleotide exchange and to allow the reaction to equilibrate. After this equilibration time and buffer exchange, the Ras proteins were concentrated and the nucleotide state was analysed by isocratic HPLC runs. By integrating the corresponding peaks for GDP and the guanosine nucleotide analogues after distinct time points, we observed that the modified nucleotides can indeed compete with GDP (*Table 1—source data 3*, *Table 1—source data 4*). Based on the relative abundance of bound nucleotides determined by the HPLC assay, the dissociation constants for the pda and bda derivatives were calculated to be 8.6±1.3 pM and 9.6±0.5 pM, respectively (overlap with the GDP elution peak prevented accurate determination in the case of the eda-derivative; Method section, *Table 1*). Thus, the attached linker has very little impact on the reversible interaction and the affinity, an important fact that must be considered for the approach of using nucleotide-competitive inhibitors (*Müller et al., 2017*). In contrast, SML-8-73-1, a GDP derivative harbouring an electrophilic group on the β-phosphate showed a dramatic loss of reversible affinity ($K_D$ = ~140 nM) (*Jeganathan et al., 2018*).

## First crystal structures of the KRasG13C mutant

To gain further insight into the binding mode of the covalently bound nucleotides, we solved the first X-ray crystal structure of the oncogenic KRasG13C mutant, in this case with covalently bound edaGDP (PDB 7ok3) and bdaGDP (PDB 7ok4) (*Figure 2*, *Figure 2—figure supplement 1*). The overall structures are very similar to the known structure of KRas:GDP (PDB 4obe), and the nucleotide scaffold, as well as the covalent linkage for both nucleotide analogues with cysteine at position 13 are well resolved in the electron density (*Figure 2C, E*, *Figure 2—figure supplement 1*). Interestingly, only the 3'-isomer of the nucleotide analogues was observed in the structures, consistent with results of the MS experiments comparing the efficiency of labelling of the dGTP derivative and the mixed isomers of the GTP derivative and showing a faster reaction for dGTP (*Figure 1C*). Both structures showed that the nucleotides are bound within the active site in a manner that is comparable to non-covalently bound GDP in other structures of Ras, with similar reversible interactions between the protein and the nucleotide and the additional well-resolved covalent link to Cys13. However, both structures lacked the $Mg^{2+}$ ions in the active site despite a $Mg^{2+}$ concentration of 2 mM in the Ras solution. The missing $Mg^{2+}$-ions are probably a result of the crystallization buffer containing $NH_4F$ or NaF, leading to precipitation of poorly soluble $MgF_2$, and this has also been observed in other PDB-deposited structures presumably because of similar effects of the reservoir solutions used in the crystallization process (e.g. PDB 4m1o, 4lyf, 4lyh, 4m21, 4m1s, 4m1t, 4m1t, 4m1y) (*Ostrem et al., 2013*). In both crystal

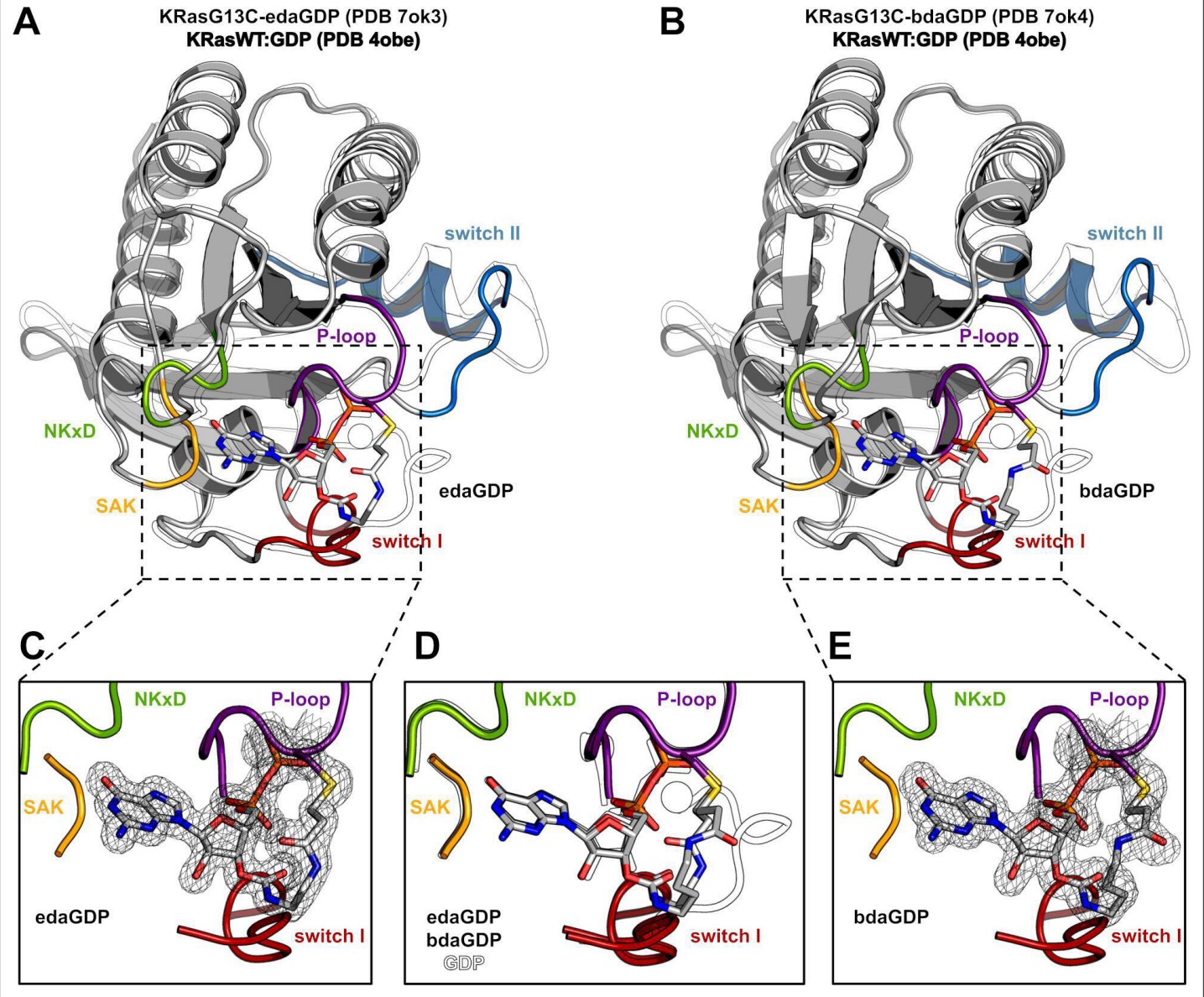

**Figure 2.** Crystal structure of KRasG13C covalently locked with either edaGDP or bdaGDP. (**A**) Comparison of KRasG13C covalently locked with edaGDP (grey, P-loop: violet, switch I: red, switch II: blue, NKxD: green, SAK: yellow, PDB 7ok3) and KRas:GDP (white, PDB 4obe). (**B**) Comparison of KRasG13C covalently locked with bdaGDP (grey, P-loop: violet, switch I: red, switch II: blue, NKxD: green, SAK: yellow, PDB 7ok4) and KRas:GDP (white, PDB 4obe). (**C**) Enlarged view of the nucleotide binding pocket in KRasG13C-edaGDP showing the 2Fo-Fc electron density map (countered at 1.0 σ). (**D**) Structural superposition of KRas:GDP and the covalently locked G13C mutants. (**E**) Enlarged view of the nucleotide binding pocket in KRasG13C-bdaGDP showing the 2Fo-Fc electron density map (countered at 1.0 σ).

The online version of this article includes the following source data and figure supplement(s) for figure 2:

**Source data 1.** Data collection and refinement statistics for KRasG13C-edaGDP and KRasG13C-bdaGDP.

**Figure supplement 1.** Structural insights into the binding mode of nucleotide-based covalent inhibitors.

structures, the eda and bda linker are remote from the Mg$^{2+}$ binding site and do not directly interfere with Mg$^{2+}$ binding, suggesting that Mg$^{2+}$ can generally bind. Thus, the structural analysis verified the nucleotide binding pose and the high reversible affinity comparable to the natural nucleotides and will guide design and optimisation of the linker and the warhead in further studies.

# Inhibition of oncogenic KRasG13C signalling by covalent nucleotide analogues

Finally, we set out to test whether the GDP nucleotide derivatives we are focusing on for cancer therapy were indeed able to inhibit oncogenic signalling by KRasG13C. In a first experiment, we tested and compared the SOS-catalysed nucleotide exchange on Ras. Whereas SOS efficiently catalysed nucleotide exchange on KRasWT, KRasG13C and as a control on KRasG13C:acetyledaGDP, it was unable to do so with the covalently locked G13C-edaGDP mutant, showing that the protein is indeed locked in the inactive state (*Figure 3A*, *Figure 3—figure supplement 1*). Interestingly, in these experiments we also observed a drastically increased intrinsic nucleotide exchange rate for KRasG13C compared to KRasWT in the absence of SOS (*Figure 3A*, blue area), an effect that probably contributes to the increased signalling and oncogenic effect of this mutant, and also this effect is abrogated in the case of covalently locked KRasG13C-edaGDP (*Hunter et al., 2015*). Similarly, no SOS-catalysed nucleotide exchange was observed in case of covalently modified KRasG13C-edaGppCp, indicating that the protein can also be trapped in the active conformation with the corresponding nucleoside-triphosphates (*Figure 3—figure supplement 1B*). Thus, the modified GppCp derivatives could also be used as artificial and irreversible activators and generally as tool compounds to further investigate the biological role of KRasG13C in cells. In addition to SOS-catalysed nucleotide exchange, GAP-stimulated GTP hydrolysis was also analysed. While the intrinsic GTP hydrolysis of KRasG13C-edaGTP ($t_{1/2}$ = 187 min) is comparable to that of KRasWT:GTP ($t_{1/2}$ = 126 min) and even faster than for the KRasG12C mutant ($t_{1/2}$ = 300 min) (*Li et al., 2021*; *Figure 3—figure supplement 2*), a drastically decreased GAP-stimulated GTP hydrolysis was observed for the G13C mutant as expected (*Scheffzek et al., 1997*).

In addition to the effects of GEFs and GAPs, we also investigated whether the covalent modification with edaGDP was sufficient to preclude effector binding. For this, we performed pull-down experiments with GST-tagged RafRBD: Whereas KRasG13C:GDP did not bind to the RafRBD, it was activated in the presence of GppNHp alone or in combination with SOS and was pulled down by RafRBD. This finding again highlights the self-activating nature of this Ras mutant, even in the absence of a stimulus (SOS). In contrast, covalently locked KRasG13C-edaGDP was neither in the presence of GppNHp nor of additional SOS able to bind to the RafRBD (*Figure 3—figure supplement 3*).

Finally, after the detailed *in vitro* characterization of the nucleotide analogues, the next experiment was to investigate whether oncogenic signalling could also be inhibited *in vivo*. Since the nucleotides are unable to cross the cell membrane without loss of the phosphate groups and full labelling of the G13C mutant was only achieved at relatively high pH values within 24 hr, we used electroporation to deliver recombinant full-length KRasWT, KRasG13C and KRasG13C-edaGDP into human cells. Importantly, we confirmed that the covalently locked protein can still be fully farnesylated *in vitro* (*Figure 3—figure supplement 4*) and the covalent modification of full-length KRasG13C was further validated through MS/MS-analysis, which revealed selective labelling of the targeted cysteine residue at position 13 (*Figure 3—figure supplement 5*). Upon electroporation into HeLa cells, a concentration-dependent increase in abundance of KRas was observed for all variants, indicating successful delivery into cells (*Figure 3B, C*, *Figure 3—figure supplements 6 and 7*). However, whereas we observed a concentration-dependent activation of the downstream signalling and upregulation of pcRaf, pAkt, pErk, and pS6 upon delivery of KRasG13C, the covalently locked variant was unable to induce these effects and, similarly to KRasWT, no increase in downstream signalling was observed (*Figure 3B, C*, *Figure 3—figure supplements 6 and 7*). In addition, upon electroporation of non-covalently modified KRasG13C:acetyledaGDP or covalently modified KRasG13C-edaGppCP, activation of the Ras pathway comparable to KRasG13C was observed, showing that covalent modification is essential for inhibition of oncogenic signalling and that artificial activation can be induced using non-hydrolysable GTP derivatives, which potentially adds further possibilities of using these nucleotide analogues to study Ras biology (*Figure 3—figure supplements 8 and 9*). Thus, our cellular data provide an additional proof-of-concept for the use of nucleotide-based covalent inhibitors and activators in KRasG13C-driven cancer.

## Conclusion

In summary, we have successfully developed nucleotide-based covalent inhibitors of oncogenic KRasG13C, a variant of KRas that is largely unexplored as a target even though it is a frequently

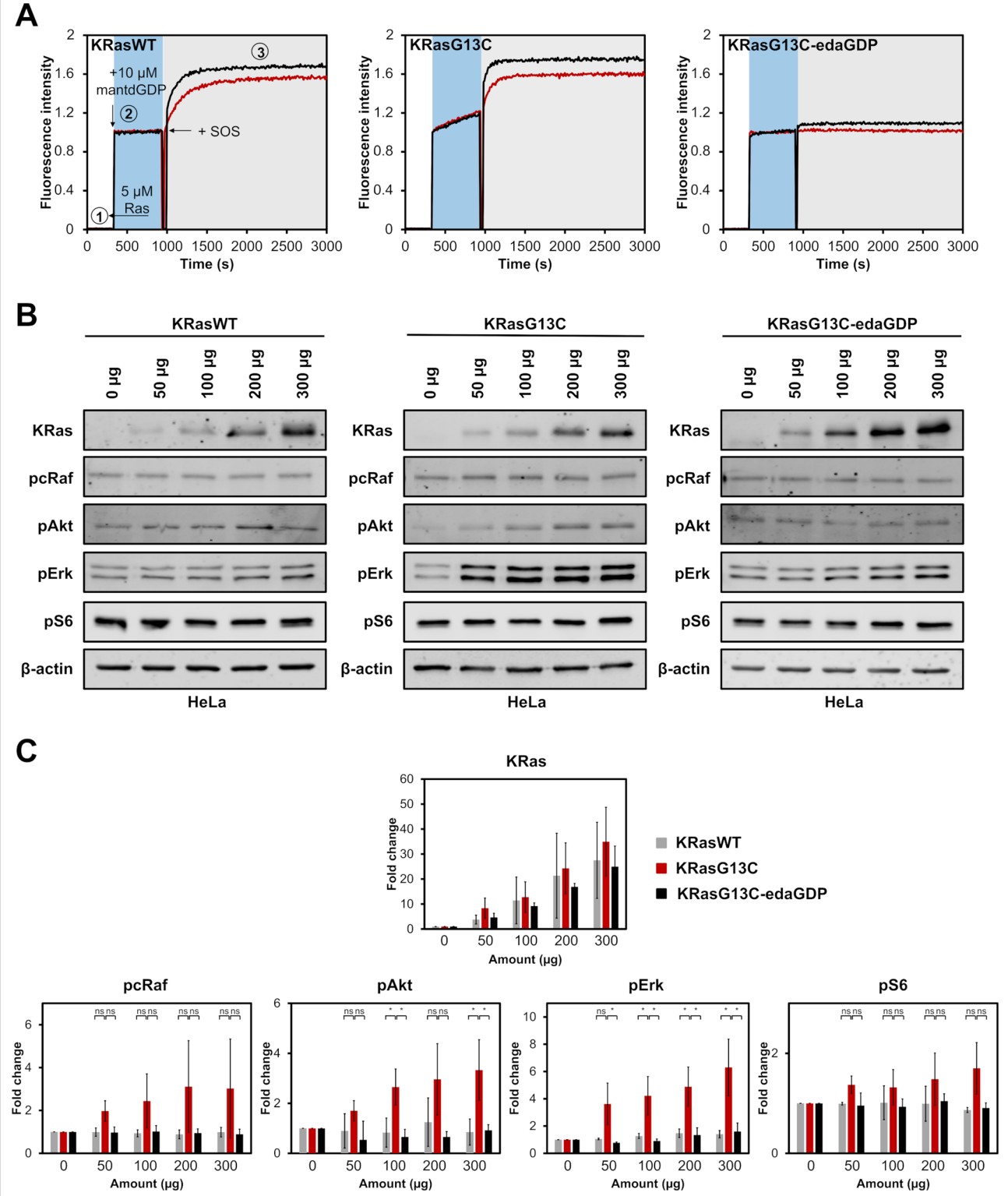

**Figure 3.** Cellular evaluation of nucleotide-based covalent inhibitors. (**A**) GEF-catalysed nucleotide exchange of KRasWT:GDP, KRasG13C:GDP and KRasG13C-edaGDP (step 1) which were mixed with an excess of mantdGDP (step 2) and subsequently with 0.25 µM (red curve) or 0.5 µM (black curve) of SOS (step 3). The intrinsic nucleotide exchange is depicted in the blue box whereas the SOS-catalysed nucleotide exchange is shown in the grey box. (**B**) Western blot analysis after electroporation of indicated amounts of full-length KRasWT, KRasG13C, and KRasG13C-edaGDP into HeLa cells. (**C**) Quantification of protein levels from western blots was performed using Empiria Studio (Li-Cor). The mean fold change was plotted

*Figure 3 continued on next page*

*Figure 3 continued*

against increasing amounts of protein. Error bars indicate the standard deviation for each measurement (n=3). One-way ANOVA was performed using GraphPad Prism. The uncropped western blots are available as source data (Goebel et al_Source_data_file_WB.xlsx).

The online version of this article includes the following source data and figure supplement(s) for figure 3:

**Source data 1.** Original uncropped Western Blots for data in *Figure 3*.

**Figure supplement 1.** GEF-catalysed nucleotide exchange.

**Figure supplement 2.** GAP-stimulated GTP hydrolysis.

**Figure supplement 3.** Effector-binding of KRasG13C is precluded upon covalent modification with edaGDP.

**Figure supplement 3—source data 1.** This is the original uncropped SDS-gel, however since it does not differ from *Figure 3—figure supplement 3*.

**Figure supplement 4.** Labelling and *in vitro* farnesylation experiments with full-length KRas.

**Figure supplement 5.** MS/MS analysis to identify the site of modification in KRasG13C.

**Figure supplement 6.** *In vitro* immunoblotting experiments.

**Figure supplement 6—source data 1.** Original uncropped Western Blots for data in *Figure 3—figure supplement 6*.

**Figure supplement 7.** Western blot analysis and quantification.

**Figure supplement 7—source data 1.** Original uncropped Western Blots for data in *Figure 3—figure supplement 7*.

**Figure supplement 8.** Electroporation of KRasG13C:acetyledaGDP.

**Figure supplement 8—source data 1.** Original uncropped Western Blots for data in *Figure 3—figure supplement 8*.

**Figure supplement 9.** Electroporation of KRasG13C-edaGppCp.

**Figure supplement 9—source data 1.** Original uncropped Western Blots for data in *Figure 3—figure supplement 9*.

observed mutation in cancer (*Visscher et al., 2016*). Thorough biochemical and structural characterization revealed that the nucleotide analogues designed and synthesized in this publication have similar affinities towards Ras compared to their natural counterparts. Since we are currently unable to directly test the nucleotides in cells, we instead extensively tested them *in vitro* regarding their ability to covalently lock KRasG13C in the inactive state and to effectively inhibit oncogenic signaling. We could show that the nucleotides prohibit (SOS-mediated) nucleotide exchange and lead to an effective interference in the induction of oncogenic effects by KRasG13C in living cells. Thus, after the first successful examples of inhibition of KRasG12C by Shokat and colleagues, our study breaks ground to effectively inhibit another important oncogenic variant of Ras by using small molecules.

Further optimization of the nucleotides to overcome the limitations regarding reactivity at physiological pH, as well as cell-permeability, are necessary and currently ongoing. In this respect, we are focusing on various linker designs based on the determined structure of the adduct, including cyclic linkers, to fine-tune the warhead's orientation and reactivity towards the cysteine at position 13. Additionally, protective esterification of the diphosphate moiety is currently being investigated to deliver prodrugs of the nucleotides into cells, which subsequently become activated by unspecific esterases (*Meier, 2017*; *Mehellou et al., 2018*). We are also investigating whether appropriate modification of the linker length and the electrophilic warhead would also make KRasG12C or other relevant mutants (e.g. HRasG12S in Costello syndrome) (*Gripp and Lin, 2012*) a possible target in a similar approach.

## Methods
### Sequence alignment

A multiple sequence alignment of the G domain Ras superfamily members was performed using (*UniProt Consortium, 2021*). The uniport accession codes that were used for the sequence alignment of the GTPases can be taken from *Supplementary file 1*.

### pKa calculations

For the protein pKa prediction, a program from OpenEye based on the Zap finite difference Poisson–Boltzmann solver was used. Regarding partial charges of the protein, Delphi radii and CHARMM36 All-Hydrogen partial charges were utilized. The linker was removed in the bda/edaGDP structures. The am1bccsym method was used to assign appropriate partial charges to the ligands. An inner dielectric of 10, an ionic strength of 0.05 M and ionization (i.e. pH) of 7.5 was applied. For cysteine,

a reference pKa of 8.6 was used. All hydrogen atoms were modeled explicitly, and except for the orientations of the OH and SH protons, which were sampled in 10° steps, the rest of the structure was static. Optimizing ionization state and SH orientation was achieved by applying ten million Monte Carlo steps (*Supplementary file 2*; *Word and Nicholls, 2011*).

## Synthesis of nucleotide-based covalent inhibitors

Synthesis of the nucleotide analogues was carried out according to the protocol established by Cremo et al. and described by *Eberth et al., 2005*; *Cremo et al., 1990*. A strong cation exchanger (Ion exchanger I, Merck, Darmstadt, Germany) was used to prepare the tributylammonium salts of the nucleotides. The commercially available sodium salt of the nucleotide (0.5 mmol) was dissolved in 3 mL ddH$_2$O and was applied on the column pre-equilibrated with pyridine/H$_2$O (1:1). Nucleotide elution was achieved using methanol/H$_2$O (1:1) and the eluate was dripped into 1 mL TBA. After monitoring the nucleotide elution by spotting samples onto a TLC plate with fluorescent indicator and removing of the methanol/H$_2$O solution the mixture was dried by repeated rotary evaporation from dry DMF (3x20 mL).

## Preparation of 5´-phosphorylimidazolidate 2',3'-O-carbonate of nucleotides

The remaining solid was dissolved in 20 mL dry DMF and CDI (2.5 mmol) was added under argon atmosphere. After stirring overnight at 4 °C to form the carbonate the reaction was quenched by the addition of absolute methanol (150 µL; *Scheme 1*).

**Scheme 1.** Preparation of 5´-phosphorylimidazolidate 2',3'-O-carbonate of nucleotides.

### HRMS (ESI-MS)

| nucleotide | calculated | [M-H]⁻ | found |
|---|---|---|---|
| GDP | 518.0232 | C$_{14}$H$_{14}$O$_{11}$N$_7$P$_2$ | 518.0220 |
| GTP | 597.9895 | C$_{14}$H$_{15}$O$_{14}$N$_7$P$_3$ | 597.9869 |
| dGTP | 650.0315 | C$_{17}$H$_{19}$O$_{13}$N$_9$P$_3$ | 650.0304 |
| GppCp* | 545.9828 | C$_{12}$H$_{15}$O$_{14}$N$_5$P$_3$ | 545.9829 |

*in case of GppCp, the cyclic carbonate was observed with a free terminal phosphate group.

## Preparation of the respective 2',3'-O-carbamates of nucleotides

The primary amine (eda, pda or bda; 2.5 mmol) dissolved in 5 mL dry DMF was slowly added to the carbonate mixture to prepare the phosphoramidate derivative. The resulting precipitate was recovered by centrifugation (10,000 rpm, 10 min) and washed three times with DMF (*Scheme 2*).

**Scheme 2.** Preparation of the respective 2',3'-O-carbamates of nucleotides.

## LCMS (ESI-MS)

| nucleotide | linker | calculated | [M-H]⁻ | found |
|---|---|---|---|---|
| GDP | eda | 578.1 | $C_{16}H_{22}N_9O_{11}P_2$ | 578.1 |
| | pda | 592.1 | $C_{17}H_{24}N_9O_{11}P_2$ | 592.1 |
| | bda | 606.1 | $C_{18}H_{26}N_9O_{11}P_2$ | 606.1 |
| GTP | eda | 658.1 | $C_{16}H_{23}N_9O_{14}P_3$ | 658.0 |
| | pda | 672.1 | $C_{17}H_{25}N_9O_{14}P_3$ | 672.1 |
| | bda | 686.1 | $C_{18}H_{27}N_9O_{14}P_3$ | 686.1 |
| dGTP | eda | 634.1 | $C_{15}H_{27}N_9O_{13}P_3$ | 634.1 |
| | pda | 662.1 | $C_{17}H_{31}N_9O_{13}P_3$ | 662.1 |
| | bda | 690.2 | $C_{19}H_{35}N_9O_{13}P_3$ | 690.2 |
| GppCp | eda | 648.1 | $C_{16}H_{29}N_9O_{13}P_3$ | 648.1 |

## Cleavage of the phosphoramidate

The solid was dissolved in 20 mL ddH$_2$O and the pH was adjusted to 1.5 with 0.25 M hydrochloric acid to hydrolyse the phosphoramidate. After stirring at 4 °C for 1–3 d the mixture was then raised to pH 7.5 using 0.25 M NaOH. The nucleotides were purified at 4 °C on a Q-Sepharose column (column volume: 130 mL) preequilibrated with 50 mM triethylammonium bicarbonate buffer (pH 7.6) and eluted by a linear gradient of 50 mM – 1 M TEAB over 600 min with a flow rate of 1 mL/min. The nucleotide containing fractions were analysed by HPLC 50 mM KPi pH 6.6, 10 mM TBAB, 16% ACN; column: ProntoSIL 120–5 C18-AQ, Bischoff, Germany and were lyophilized several times from ddH$_2$O to remove the buffer (*Scheme 3*).

**Scheme 3.** Cleavage of the phosphoramidate.

## HRMS (ESI-MS)

| nucleotide | linker | calculated | [M-H]⁻ | found | purity (%) | yield (%) |
|---|---|---|---|---|---|---|
| GDP | eda | 528.0651 | $C_{13}H_{20}O_{12}N_7P_2$ | 528.0654 | 90 | 38 |
| | pda | 542.0802 | $C_{14}H_{22}O_{12}N_7P_2$ | 542.0809 | >95 | 65 |
| | bda | 556.0964 | $C_{15}H_{24}O_{12}N_7P_2$ | 556.0960 | >95 | 29 |
| GTP | eda | 608.0314 | $C_{13}H_{21}O_{15}N_7P_3$ | 608.0289 | 89 | 56 |
| | pda | 662.0471 | $C_{14}H_{23}O_{15}N_7P_3$ | 662.0471 | 87 | 83 |
| | bda | 636.0621 | $C_{15}H_{25}N_7O_{15}P_3$ | 636.0603 | 94 | 67 |
| dGTP | eda | 592.0359 | $C_{13}H_{21}O_{14}N_7P_3$ | 592.0349 | 84 | 23 |
| | pda | 606.0516 | $C_{14}H_{23}O_{14}N_7P_3$ | 606.0521 | 93 | 11 |
| | bda | 620.0672 | $C_{15}H_{25}N_7O_{14}P_3$ | 620.0663 | 90 | 34 |
| GppCp | eda | 606.0516 | $C_{14}H_{23}N_7O_{14}P_3$ | 606.0513 | 89 | - |

## Insertion of the Michael acceptor or non-reactive acetamido derivative

The nucleotide based covalent inhibitors were prepared by dissolving eda, pda or bda nucleotides in a small amount of tetraborate buffer (100 mM, pH 8.5) and adding N-acryloxysuccinimide (1 eq.) dissolved in 25 µL DMSO. The reaction mixture was stirred for 24 h at room temperature and the reaction progress was monitored using HPLC. The covalent nucleotide analogues were purified using Q-Sepharose as described above and were stored at –20 °C as concentrated solutions (~100 mM) in 200 mM HEPES (pH 7.5; *Scheme 4*).

**Scheme 4.** Insertion of the Michael acceptor or non-reactive acetamido derivative.

## HRMS (ESI-MS)

| Acrylamides | linker | calculated | [M-H]⁻ | found | purity (%) | yield (%) |
|---|---|---|---|---|---|---|
| GDP | eda | 582.0751 | $C_{16}H_{22}N_7O_{13}P_2$ | 582.0741 | >95 | 57 |
| | pda | 596.0907 | $C_{17}H_{24}N_7O_{13}P_2$ | 596.0907 | 87 | 65 |
| | bda | 610.1064 | $C_{18}H_{26}N_7O_{13}P_2$ | 610.1062 | >95 | 78 |
| GTP | eda | 662.0414 | $C_{16}H_{23}N_7O_{16}P_3$ | 662.0394 | 93 | 54 |
| | pda | 676.0571 | $C_{17}H_{25}N_7O_{16}P_3$ | 676.0551 | >95 | 63 |
| | bda | 690.0727 | $C_{18}H_{27}N_7O_{16}P_3$ | 690.0705 | 86 | 75 |
| dGTP | eda | 646.0465 | $C_{16}H_{23}N_7O_{15}P_3$ | 646.0452 | 93 | 63 |
| | pda | 660.0621 | $C_{17}H_{25}N_7O_{15}P_3$ | 660.0611 | 90 | 81 |
| | bda | 674.0778 | $C_{18}H_{27}N_7O_{15}P_3$ | 674.0771 | 93 | 74 |
| GppCp | eda | 660.0621 | $C_{17}H_{27}N_7O_{16}P_3$ | 660.0619 | 79 | 15* |
| Acetylamides | linker | calculated | [M-H]⁻ | found | purity (%) | yield (%) |
| GDP | eda | 570.0751 | $C_{15}H_{22}N_7O_{13}P_2$ | 570.0734 | >95 | 57 |
| | pda | 584.0907 | $C_{16}H_{24}N_7O_{13}P_2$ | 584.0889 | 90 | 51 |
| | bda | 598.1064 | $C_{17}H_{26}N_7O_{13}P_2$ | 598.1042 | 79 | 73 |
| GTP | eda | 650.0414 | $C_{15}H_{23}N_7O_{16}P_3$ | 650.0392 | >95 | 23 |
| | pda | 664.0571 | $C_{16}H_{25}N_7O_{16}P_3$ | 664.0553 | 94 | 40 |
| | bda | 678.0727 | $C_{17}H_{27}N_7O_{16}P_3$ | 678.0700 | 93 | 31 |

*in case of GppCp, the total yield was calculated over 4 steps.

## Protein expression and purification

KRasWT$_{1-169}$ and KRasG13C$_{1-169}$ Cys-light (C51S C80L C118S) were expressed in BL21 (DE3) *E. coli* whereas the full-length KRasWT and KRasG13C were expressed in BL21 (DE3) RIL *E. coli* at 37 °C. Protein expression was induced at A600 nm of 0.5 by addition of 0.2–0.3 mM isopropyl-b-D-thiogalactoside (IPTG), and growth was continued at 19 °C overnight. The bacteria were collected by centrifugation and the obtained pellet resuspended in Ni-NTA buffer (KRas$_{1-169}$: 50 mM Tris pH 8.0, 250 mM NaCl, 40 mM imidazole, 4 mM MgCl$_2$, 10 µM GDP, 1 mM dithiothreitol (DTT), and 5% glycerol; full-length KRas: 50 mM HEPES pH 7.2, 500 mM LiCl, 2 mM MgCl$_2$, 10 µM GDP, 2 mM β-mercaptoethanol (βME)). The cells were lysed with a microfluidizer, and after addition of protease inhibitor cocktail (Roche complete EDTA free) and 1% CHAPS (w/v) stirring was continued for 1 hr at 4 °C. The lysate was cleared by centrifugation (35,000 x *g*, 1 h) and the supernatant was loaded onto a Ni-affinity chromatography column (Qiagen Ni-NTA Superflow, 20 mL) pre-equilibrated with Ni-NTA buffer. KRas$_{1-169}$ proteins were eluted with a linear gradient of imidazole buffer (40 mM – 500 mM), whereas the full-length KRas proteins were collected with a stepwise elution (2, 5, 10, 20, 30, 50, and 100% of Ni-NTA-buffer containing 500 mM imidazole). For cleavage of the N-terminal hexahistidine-tag, TEV protease was added to the pooled elution fractions and dialyzed overnight into dialysis buffer at 4 °C (KRas$_{1-169}$: 25 mM Tris pH 8.0, 100 mM NaCl, 4 mM MgCl$_2$, 10 µM GDP, 1 mM DTT, and 5% glycerol; full-length KRas: 20 mM HEPES pH 7.2, 200 mM NaCl, 2 mM MgCl$_2$, 10 µM GDP, 2 mM βME and 5% glycerol). The cleaved protein was then applied to a reverse Ni-affinity chromatography column and finally purified by size-exclusion chromatography (GE HiLoad 16/60 Superdex 75 pg) in a final buffer containing 20 mM HEPES pH 7.5, 100 mM NaCl, 2 mM MgCl$_2$, 10 µM GDP, 1 mM TCEP, and 5% glycerol. GST-tagged RafRBD was purified as described previously by affinity chromatography and subsequent size exclusion chromatography in a final buffer containing 20 mM HEPES pH 7.5, 100 mM NaCl, 2 mM MgCl$_2$, 1 mM TCEP, and 5% glycerol (*Herrmann et al., 1995*).

### Nucleotide exchange

A total of 50 µM Ras protein was incubated with a 10-fold excess of nucleotides in 20 mM HEPES (pH 7.5), 100 mM NaCl, 2 mM MgCl$_2$, 1 mM TCEP, 5% glycerol, and 10 mM EDTA for 3 hr at 4 °C. Nucleotide exchange was terminated by the addition of 20 mM MgCl$_2$ and Ras proteins were washed using centrifugal filter devices to remove any unbound nucleotide. Nucleotide exchange was controlled by isocratic HPLC runs.

### Covalent modification of proteins

To analyse the amount of covalent modification of KRasG13C$_{1-169}$, 50 µM Ras protein was incubated with a 10-fold excess of the acryl-bearing nucleotides in 100 mM CHES (pH 9.5), 50 mM NaCl, 1 mM TCEP, and 1 mM EDTA. After incubation at room temperature for the appropriate time, the modification of the protein was controlled by ESI-MS. For covalent modification of full-length KRasG13C, a nucleotide exchange with a 10-fold excess of the acryl-bearing nucleotides at pH 7.5 was first performed following incubation at room temperature for 24 h at pH 9.5 for covalent protein modification. Covalent protein modification was controlled by ESI-MS. The MS spectra were recorded on a VelosPro IonTrap (Thermo Scientific) with an EC 50/3 Nucleodur C18 1.8 µm column (Macherey and Nagel) and a gradient of the mobile phase A (0.1% formic acid in water) to B (0.1% formic acid in acetonitrile).

### Stopped-flow experiments

The association kinetics of the nucleotide analogues were analyzed with a SX-20 stopped-flow instrument (Applied Photophysics) at 25 °C in a buffer consisting of 25 mM HEPES (pH 7.5), 100 mM NaCl, 1 mM MgCl$_2$, and 0.5 mM TCEP. 1 µM of nucleotide-free KRas (*Müller et al., 2017*) in one syringe was mixed rapidly with 2 µM mantdGDP in the other. In subsequent experiments, the second syringe contained competing nucleotides (1, 2, and 6 µM) in addition to 2 µM mantdGDP. The resulting progress curves were globally fit using KinTek Explorer to obtain the corresponding association rate constants as previously described (*Table 1—source data 1*, *Table 1—source data 2 Johnson et al., 2009*).

### HPLC-based approach for determination of affinities relative to GDP

The relative affinities of the nucleotide analogues were measured using an HPLC-based approach. 50 µM of KRasWT$_{1-169}$:GDP was mixed with 50 µM of the acetamide derivatives and incubated either for 7 days at room temperature in the absence of EDTA, for 24 hr at 4 °C in the presence of 10 mM EDTA, or for 1 hr at room temperature in the presence of SOS in a buffer consisting of 20 mM HEPES (pH 7.5), 100 mM NaCl, 2 mM MgCl$_2$, 1 mM TCEP and 5% glycerol. After incubation, the Ras proteins were washed five times with buffer (15 mL) using centrifugal filter devices to remove any unbound nucleotide, concentrated and analysed by isocratic HPLC runs. The resulting curves were analysed by integrating the corresponding peaks for GDP and the guanosine nucleotide analogues after distinct time points using the Agilent ChemStation Software (*Table 1—source data 3*, *Table 1—source data 4*).

### Calculation of the K$_D$ values of the nucleotide analogues

Based on the percentage distribution determined by the HPLC assay, the relative association constants (K$_{relA}$) for pdaGDP and bdaGDP could subsequently be calculated. In contrast, the superposition of the GDP signal prevented an exact determination of the relative association constant for edaGDP. K$_{relA}$ values of 0.34 and 0.28 relative to GDP were determined for pdaGDP and bdaGDP, respectively. Finally, considering the K$_D$ value of 2.5 pM for GDP described by Jeganathan et al. the corresponding dissociation constant K$_D$ values could be determined, which are shown in *Table 1—source data 4*; *Jeganathan et al., 2018*.

$$Ras + GDP \xrightarrow{k_{GDP}} Ras:GDP \tag{1a}$$

$$Ras + aGDP \xrightarrow{k_{aGDP}} Ras:aGDP \tag{1b}$$

By rearranging the equilibrium reactions listed in (*Equation 1a*) and (*Equation 1b*), the following relationships are obtained for K$_{GDP}$ (*Equation 2a*) and K$_{aGDP}$ (*Equation 2b*):

$$K_{GDP} = \frac{[Ras:GDP]}{[Ras][GDP]} \tag{2a}$$

$$K_{aGDP} = \frac{[Ras:aGDP]}{[Ras][aGDP]} \tag{2b}$$

Assuming that the concentration for aGDP used in the HPLC assay is 50 µM, the following GDP concentrations for pdaGDP (*Equation 3a*) and bdaGDP (*Equation 3b*) can be determined at time t=0 hr using the percentages determined in *Table 1—source data 3*:

$$GDP_{total} = \frac{\%_{GDP}}{\%_{pdaGDP}} = \frac{37\%}{63\%} \times 50\mu M = 29.4\mu M \tag{3a}$$

$$GDP_{total} = \frac{\%_{GDP}}{\%_{bdaGDP}} = \frac{43\%}{57\%} \times 50\mu M = 37.7\mu M \tag{3b}$$

The concentrations of the components listed in equation (*Equation 2b*) can be calculated as follows for pdaGDP (*Equation 4a*, *Equation 4b*, *Equation 4c*, *Equation 4d*) and bdaGDP (*Equation 5a*, *Equation 5b*, *Equation 5c*, *Equation 5d*) by considering the total GDP concentrations calculated previously:

$$[Ras:pdaGDP] = \frac{47}{100} \times 29.4\mu M = 13.8\mu M \tag{4a}$$

$$[Ras:GDP] = \frac{53}{100} \times 29.4\mu M = 15.6\mu M \tag{4b}$$

$$[pdaGDP] = 50\mu M - 13.8\mu M = 36.2\mu M \tag{4c}$$

$$[GDP] = 29.4\mu M - 15.6\mu M = 13.8\mu M \tag{4d}$$

$$[Ras:bdaGDP] = \frac{39.5}{100} \times 37.7\mu M = 14.9\mu M \tag{5a}$$

$$[Ras:GDP] = \frac{60.5}{100} \times 37.7\mu M = 22.8\mu M \tag{5b}$$

$$[bdaGDP] = 50\mu M - 14.9\mu M = 35.1\mu M \tag{5c}$$

$$[GDP] = 37.7\mu M - 22.8\mu M = 14.9\mu M \tag{5d}$$

The relative association constant $K_{relA}$ can be calculated by the following formula (*Equation 6*):

$$K_{relA} = \frac{K_{aGDP}}{K_{GDP}} = \frac{[Ras:aGDP] \times [GDP]}{[aGDP] \times [Ras:GDP]} \tag{6}$$

By inserting the concentrations of each component determined from (*Equation 4a*, *Equation 4b*, *Equation 4c*, *Equation 4d*) and (*Equation 5a*, *Equation 5b*, *Equation 5c*, *Equation 5d*) into (*Equation 6*), the relative association constants $K_{relA}$ for pdaGDP (*Equation 7a*) and bdaGDP (*Equation 7b*) can be calculated:

$$K_{relA} = \frac{K_{pdaGDP}}{K_{GDP}} = \frac{13.8\mu M \times 13.8\mu M}{36.2\mu M \times 15.6\mu M} = 0.34 \tag{7a}$$

$$K_{relA} = \frac{K_{bdaGDP}}{K_{GDP}} = \frac{14.9\mu M \times 14.9\mu M}{35.1\mu M \times 22.8\mu M} = 0.28 \tag{7b}$$

Based on the relative association constants $K_{relA}$, the $K_A$ values of the nucleotide analogues can be calculated by multiplication with the association constant of GDP:

$$K_A = K_A GDP \times K_{relA} \tag{8}$$

The $K_D$ value of 2.5 pM described by *Jeganathan et al., 2018* was used as the reference value for GDP. By substituting the reference value into $K_A = 1/K_D$, a $K_A$ value for GDP of 0.4 pM$^{-1}$ was determined. Accordingly, the $K_A$ values for pdaGDP (*Equation 9a*) and bdaGDP (*Equation 9b*) are as follows:

$$K_A pdaGDP = 0.4pM^{-1} \times 0.34 = 0.136pM^{-1} \tag{9a}$$

$$K_A bdaGDP = 0.4pM^{-1} \times 0.28 = 0.112pM^{-1} \tag{9b}$$

By converting the determined $K_A$ values into the corresponding $K_D$ values using $K_D = 1/K_A$, the following dissociation constants could be determined for pdaGDP (*Equation 10a*) and bdaGDP (*Equation 10b*):

$$K_D pdaGDP = \frac{1}{0.136pM^{-1}} = 7.4pM \qquad (10a)$$

$$K_D bdaGDP = \frac{1}{0.112pM^{-1}} = 8.9pM \qquad (10b)$$

## Crystallization and structure determination

KRasG13C$_{1-169}$ Cys-light (C51S C80L C118S) was covalently modified by incubating 100 µM KRas with a 10-fold excess of the acryl-bearing nucleotide in 100 mM CHES (pH 9.5), 50 mM NaCl, 1 mM TCEP, and 1 mM EDTA at room temperature for 24 hr. Modification of the protein was monitored by ESI-MS and terminated by the addition of 20 mM MgCl$_2$. The protein was purified by size-exclusion chromatography (GE HiLoad 16/60 Superdex 75 pg) in a final buffer containing 20 mM HEPES pH 7.5, 100 mM NaCl, 2 mM MgCl$_2$, 1 mM TCEP, and 5% glycerol, and subsequently concentrated to 67 mg/mL. To identify the initial crystallization conditions, commercially available protein crystallization screens (JCSG Core I –IV Suites, PEGs and PACT) were used. Using a TTP labtech Mosquito LCP crystal liquid-handling robot, 100 nL of protein solution was mixed with 100 nL reservoir solution in 96-well plates, and crystals were grown using the hanging-drop method at 20 °C. After 1 day of incubation, one successful crystallization condition was obtained for KRasG13C-edaGDP (0.2 M (NH$_4$) F, 20% PEG3350) and KRasG13C-bdaGDP (0.2 M NaF, 20% PEG3350), and the crystals were cryoprotected in mother liquor and flash cooled in liquid nitrogen. The data sets were collected at the PXII X10SA beamline of the Swiss Light Source (Paul Scherrer Institute, Villigen, Switzerland) and indexed and scaled using XDS (*Kabsch, 2010*). The crystal structures were solved by molecular replacement with PHASER using PDB 4obe as a template (*Read, 2001*). The manual modification of the molecule of the asymmetric unit was performed using the program COOT (*Emsley and Cowtan, 2004*), and with the help of the Dundee PRODRG server (*Schüttelkopf and van Aalten, 2004*), the inhibitor topology file was generated. For multiple cycles of refinement, PHENIX.refine (*Adams et al., 2010*) was employed, the final structure was evaluated by the PDB_REDOserver (*Joosten et al., 2014*) and crystal structures were visualized using PyMOL. Data collection, structure refinement statistics, and further details for data collection are provided in *Figure 2—figure supplement 1*.

## Guanine nucleotide-exchange factor assay

SOS-catalysed nucleotide exchange was monitored at 25 °C in a FluoroMax-3 spectrofluorometer (excitation at 360 nm, emission at 440 nm) in 20 mM HEPES (pH 7.5), 100 mM NaCl, 2 mM MgCl$_2$, and 1 mM TCEP. 5 µM KRas$_{1-169}$ was mixed with 10 µM mantdGDP and subsequently with different concentrations of SOS (0.25 µM and 0.5 µM) (*Figure 3A*; *Figure 3—figure supplement 1*).

## GAP-stimulated GTP hydrolysis

First, for KRasWT$_{1-169}$, nucleotide exchange with GTP was performed at pH 7.5, and for KRasG13C$_{1-169}$, covalent modification with a 10-fold excess of acryl-edaGTP at pH 9.5 was done, both in the presence of 50 mM EDTA to block intrinsic GTP hydrolysis. After incubation, the Ras proteins were washed five times with buffer (20 mM HEPES (pH 7.5), 100 mM NaCl, and 1 mM TCEP) using centrifugal filter devices to remove any unbound nucleotide. Nucleotide exchange of KRasWT$_{1-169}$ was controlled by isocratic HPLC runs, and covalent modification of KRasG13C$_{1-169}$ was verified by ESI-MS. GTP hydrolysis of Ras proteins was initiated by addition of 2 mM MgCl$_2$ in the absence or presence of Ras-GAP (1:1000 for KRasWT$_{1-169}$:GTP and 1:1 for KRasG13C$_{1-169}$-edaGTP). At defined time points (0, 5, 10, 15, 20, 30, 45, 60, 90 and 120 min), samples were taken and immediately snap-frozen in liquid nitrogen. For KRasWT$_{1-169}$:GTP, samples were thawed and denatured for 5 min at 95 °C. Samples were centrifuged at 14,000 rpm for 10 min at 4 °C and subsequently analysed by isocratic HPLC runs. For KRasG13C$_{1-169}$-edaGTP, samples were centrifuged and analysed by ESI-MS. The relative amount of each nucleotide was determined by integrating the area of the GTP and GDP peaks using Origin (*Figure 3—figure supplement 2*; *Eberth and Ahmadian, 2009*).

## Effector binding (pull-down experiments)

Pull-down experiments were performed in 20 mM HEPES pH 7.5, 50 mM NaCl and 2 mM MgCl$_2$. 10 µg KRasG13C:GDP or the covalently modified KRasG13C-edaGDP and 20 µg GST-tagged cRafRBD (amino acids 51–131) were incubated with/ without 100 µM GppNHp and with/ without 1 µg SOS over night at room temperature. Afterwards, 50 µL glutathione magnetic beads were added to each

sample for 30 min. After washing the beads with 500 µL buffer, the beads were settled with a magnet and the supernatant was carefully removed. The beads were resuspended in 50 µL of 4xSDS-loading buffer and visualized via SDS-PAGE (*Figure 3—figure supplement 3*).

## *In vitro* farnesylation

50 µM of full-length KRas protein was mixed with 250 µM farnesyl pyrophosphate (FPP) and 10 µM farnesyltransferase (FTase). After incubation at room temperature for 1 hr the mixture was centrifugated at 14,000 rpm for 10 min at 4 °C and analyzed via ESI-MS (*Figure 3—figure supplement 4*).

## Competition assay

To analyze the amount of covalent modification of KRasG13C$_{1-169}$ (Cys-light) with the acryl-bearing nucleotide analogue (edaGDP) under competitive conditions with GDP and GTP, four different conditions were tested (buffer: 100 mM CHES pH 9.5., 50 mM NaCl, 1 mM TCEP, 2 mM MgCl$_2$, 5% glycerol). 5 µM KRasG13C (Cys-light) was incubated (1) in the presence of 36 µM edaGDP with and without SOS, (2) in the presence of 305 µM edaGDP, 36 µM GDP and 305 µM GTP with and without SOS, (3) in the presence of 36 µM edaGDP, 36 µM GDP and 305 µM GTP with and without SOS and (4) 36 µM edaGDP, 360 µM GDP, and 3050 µM GTP with and without SOS. After incubation for 1, 2, 3, 4, 5, 22, and 24 hr at room temperature, the covalent protein modification was controlled by ESI-MS.

## MS/MS analysis

For MS/MS analysis, the samples were dissolved in 100 mM TEAB and incubated for 1 hr at 55 °C in the presence of 10 mM TCEP. 17 mM iodoacetamide was added and the samples were incubated in the dark at room temperature for 30 min. Samples were precipitated by adding pre-chilled acetone and stored overnight at –20 °C. After drying the pellets, Trypsin (Roche) was added and the samples were digested at 37 °C with 300 rpm shaking overnight. The digestion was quenched by the addition of 2% TFA and a stage tip purification (*Rappsilber et al., 2007*) was performed, samples were evaporated to dryness and stored at –20 °C until MS/MS analysis. For nanoHPLC-MS/MS analysis samples were dissolved in 20 µL of 0.1% TFA in water and 3 µL were injected onto an UltiMateTM 3000 RSLCnano system (ThermoFisher Scientific, Germany) online coupled to a Q Exactive Plus Hybrid Quadrupole-Orbitrap Mass Spectrometer equipped with a nanospray source (Nanospray Flex Ion Source, Thermo Scientific). All solvents were LC-MS grade. Samples were injected onto a pre-column cartridge (5 µm, 100 Å, 300 µm ID * 5 mm, Dionex, Germany) using 0.1% TFA in water as eluent with a flow rate of 30 µL/min. Desalting was performed for 5 min with eluent flow to waste followed by back-flushing of the sample during the whole analysis from the pre-column to the PepMap100 RSLC C18 nano-HPLC column (2 µm, 100 Å, 75 µm ID ×50 cm, nanoViper, Dionex, Germany) using a linear gradient starting with 95% solvent A (water containing 0.1% formic acid) / 5% solvent B (acetonitrile containing 0.1% formic acid) and increasing to 30% solvent B in 90 min using a flow rate of 300 nL/min. Afterwards, the column was washed (two steps 60 and 95% solvent B) and re-equilibrated to starting conditions. The nanoHPLC was online coupled to the Quadrupole-Orbitrap Mass Spectrometer using a standard coated SilicaTip (ID 20 µm, Tip-ID 10 µM, New Objective, Woburn, MA, USA). Mass range of m/z 300–1,650 was acquired with a resolution of 70000 for a full scan, followed by up to 10 high energy collision dissociation (HCD) MS / MS scans of the most intense at least doubly charged ions using a resolution of 17500 and a NCE energy of 25%. Data evaluation was performed using MaxQuant software (*Cox and Mann, 2008*) (v.1.6.3.4) including the Andromeda search algorithm (*Cox et al., 2011*) and searching the KRas sequence together with a database containing typical contaminants like keratins, trypsin etc., which is included in the MaxQuant software. The search was performed for full enzymatic trypsin cleavages allowing two miscleavages. For database search oxidation of methionine and N-terminal acetylation of proteins, carbamidomethylation of cysteines, and artificial modification of cysteines were defined as variable modifications. The mass accuracy for full mass spectra was set to 20 ppm (first search) and 4.5 ppm (second search), respectively and for MS/MS spectra to 20 ppm. The false discovery rates for peptide and protein identification were set to 1%. For further analysis, the peptide intensities of KRas were compared for modified and unmodified KRas. (*Figure 3—figure supplement 5*).

## Cell culture

HeLa cells were obtained from the American Type Culture Collection (ATCC) and were cultured in DMEM medium (Gibco) supplemented with 10% fetal bovine serum (FBS) (PAN-Biotech) and 1% penicillin/streptomycin (Gibco). Cells were cultured in a humidified incubator at 37 °C in the presence of 5% $CO_2$. The identity of the cells was validated by STR profiling and the cells were tested negative for mycoplasma contamination.

## Electroporation

Electroporation of full-length KRas constructs (KRasWT, KRasG13C, KRasG13C-edaGDP, KRasG13C:acetyledaGDP and KRasG13C-edaGppCp) was performed based on the protocol described by *Alex et al., 2019* using the Neon Transfection System Kit (Thermo Fisher). For electroporation, 3 million cells per experiment were harvested by trypsinization, washed with PBS, and resuspended in 85 µL of the electroporation buffer R (Thermo Fisher). Increasing amounts of recombinant protein samples for each construct (0, 50, 100, 200 and 300 µg) were diluted 1:1 in buffer R followed by the addition of 30 µL of this protein master mix to the cell suspension. This cellular slurry was loaded into a 100 mL Neon Pipette Tip (Thermo Fisher) and electroporated with 2x35 ms pulses at 1000 V. After electroporation, the cells were washed twice with PBS (15 mL) to remove non-internalized extracellular protein and the cell pellet was resuspended in 2 mL complete growth media. Cells were transferred into six-well tissue culture plates (Sarstedt) and incubated for 24 hr at 37 °C and 5% $CO_2$ in a humidified incubator for recovery before being processed for western blotting analysis.

## Western blot analysis

After recovery of the electroporation, cells were washed twice with ice-cold PBS and lysed in 100 µL of phosphatase and protease inhibitor containing RIPA buffer (Cell Signaling Technology). Cells were incubated on ice for 30 min and then harvested by scraping followed by centrifugation at 14,000 rpm for 10 min at 4 °C. Protein concentrations were determined using the Pierce BCA protein assay (Thermo) following the manufacturer's recommended procedure. Equal amounts of protein (10 µg) were analyzed by SDS-PAGE and transferred to Immobilon-FL PVDF membranes (Merck Millipore) using Pierce 1-step transfer buffer (Thermo) and the Pierce Power Blotter (Thermo). Membranes were washed with $ddH_2O$ for 5 min, blocked with OdysseyBlocking Buffer TBS (Li-Cor) for 1 hr at room temperature and then incubated with primary antibodies diluted in OdysseyBlocking Buffer TBS overnight at 4 °C with gentle agitation. KRas (Sigma Aldrich, SAB1404011-100UG), pcRafS338 (CST, 9427), tAkt1 (CST, 2938), pAktS473 (CST, 4060), tErk (CST, 4696), pErkT202/Y204 (CST, 4370), pS6S235/236 (CST, 4858), and β-actin (CST, 4970/Sigma-Aldrich, A5441) antibodies were used to detect the individual proteins. After primary antibody incubation, membranes were washed three times with TBS-T (50 mM Tris, 150 mM NaCl, 0.05% Tween 20, pH 7.4) for 5 min before being incubated with secondary antibodies (anti-mouse IgG (H+L) (DyLight 680 Conjugate) (CST, 5470) / anti-rabbit IgG (H+L) (DyLight 800 4 X PEG Conjugate) (CST, 5151)) diluted in OdysseyBlocking Buffer TBS for 1 hr at room temperature with gentle agitation. After secondary antibody incubation, the membranes were washed three times for 5 min with TBS-T and then scanned using an OdysseyCLx imaging system (Li-Cor). Quantification of protein levels from western blots was performed using Empiria Studio (Li-Cor) (*Figure 3B, C*, *Figure 3—figure supplements 6–9*).

## Acknowledgements

This work was co-funded by the Deutsche Forschungsgemeinschaft (DFG; RA 1055/5–1 and GO 284/10–1) and the Drug Discovery Hub Dortmund (DDHD). We thank Nathalie Bleimling, Andreas Arndt and Paul Siebers for invaluable technical assistance as well as Prof. Christian Herrmann for providing the plasmid for c-RafRBD. We acknowledge the Paul Scherrer Institut, Villigen, Switzerland for provision of synchrotron radiation beamtime at beamline X10SA of the SLS and would like to thank the staff of the SLS for assistance. We acknowledge financial support by Deutsche Forschungsgemeinschaft and Technische Universität Dortmund/TU Dortmund University within the funding programme Open Access Costs.

## Additional information

### Funding

| Funder | Grant reference number | Author |
|---|---|---|
| Deutsche Forschungsgemeinschaft | RA 1055/5-1 | Roger S Goody |
| Deutsche Forschungsgemeinschaft | GO 284/10-1 | Daniel Rauh |
| TU Dortmund University | | Matthias P Müller |

The funders had no role in study design, data collection and interpretation, or the decision to submit the work for publication.

### Author contributions

Lisa Goebel, Validation, Investigation, Visualization, Methodology, Writing - original draft; Tonia Kirschner, Sandra Koska, Amrita Rai, Petra Janning, Helge Vatheuer, Investigation; Stefano Maffini, Resources, Methodology; Paul Czodrowski, Supervision, Methodology; Roger S Goody, Matthias P Müller, Daniel Rauh, Conceptualization, Supervision, Funding acquisition, Project administration, Writing – review and editing

### Author ORCIDs

Lisa Goebel ⓘ http://orcid.org/0000-0003-3477-1228
Amrita Rai ⓘ http://orcid.org/0000-0002-5471-5250
Stefano Maffini ⓘ http://orcid.org/0000-0001-6380-6560
Roger S Goody ⓘ http://orcid.org/0000-0002-0772-0444
Matthias P Müller ⓘ http://orcid.org/0000-0002-1529-8933
Daniel Rauh ⓘ http://orcid.org/0000-0002-1970-7642

### Decision letter and Author response

Decision letter https://doi.org/10.7554/eLife.82184.sa1
Author response https://doi.org/10.7554/eLife.82184.sa2

## Additional files

### Supplementary files

• Supplementary file 1. Sequence alignment of Ras small GTPase superfamily. Multiple sequence alignment of the G domain Ras superfamily members using Uniprot (the uniport accession codes can be taken from this figure). Based on the marked areas that are shown in *Figure 1—figure supplement 1*, cysteines that are within the reach of the warhead of the eda linker are highlighted in gray. In contrast, regions that are also within reach of the bda linker are highlighted in blue. The cysteines located within these regions are marked in red. Only ~7% of the GTPases have cysteines within the P-loop and are therefore potential off-targets of the designed nucleotide derivatives.

• Supplementary file 2. pKa calculations. Overview of calculated pKa values of the KRasG12C and G13C mutants in the presence (+) and absence (-) of GDP. The linker in case of the G13C mutants was removed so that all featured structures had GDP and a free cysteine in its active center. The pKa calculations showed that both cysteines at position 12 and 13 have similar pKa values indicating that also position 13 should generally be addressable by covalent warheads.

• MDAR checklist

• Source data 1. Summary of all uncropped western blots used in Figure 3 and the Figure supplements.

### Data availability

Diffraction data have been deposited in PDB under the accession code 7ok3 and 7ok4.

The following datasets were generated:

| Author(s) | Year | Dataset title | Dataset URL | Database and Identifier |
|---|---|---|---|---|
| Goebel L, Mueller MP, Rauh D | 2022 | Crystal Structure of KRasG13C in Complex with Nucleotide-based covalent Inhibitor bdaGDP | https://www.rcsb.org/structure/7OK4 | RCSB Protein Data Bank, 7OK4 |
| Goebel L, Mueller MP, Rauh D | 2022 | Crystal Structure of KRasG13C in Complex with Nucleotide-based covalent Inhibitor edaGDP | https://www.rcsb.org/structure/7OK3 | RCSB Protein Data Bank, 7OK3 |

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
