## [Editor Report]

The authors present important information regarding the possibility of targeting the oncogenic K-Ras(G13C) mutant with nucleotide competitors. The experiments represent a solid support of the claims and show that this approach can work despite concerns about the high affinity of GTP and its high cellular concentration. These results will be of high interest for all working in the Ras field and in targeting oncogenes with small molecules. A weakness of the manuscript is the lack of direct physiological insights.

---

## [Decision Letter]

**Decision letter after peer review:**

Thank you for submitting your article "Targeting oncogenic KRasG13C with nucleotide-based covalent inhibitors" for consideration by *eLife*. Your article has been reviewed by 3 peer reviewers, and the evaluation has been overseen by a Reviewing Editor and Volker Dötsch as the Senior Editor. The reviewers have opted to remain anonymous.

Essential revisions:

1) Can the edaGDP derivative competes with 1 mM of GDP and GTP in covalently modifying KRAS G13C and the wild type KRAS (well to a lesser extent)?I see the authors also employed Cys light version of RAS like the Shokat group in some of their assays and perhaps wild type protein is probably better here as well as anyway they are performing MS analysis.

2) Related: K-Ras wildtype was labeled by eda/pda/bdaGDP to a much lower extent (Figure 1D, E). however, K-Ras(G12C) Cyslight was not labeled by these GDP analogs at all. Do the authors have any MS2 data to identify the site of modification in the case of K-Ras WT? There is growing interest in covalent adducts to RAS, so identifying the nearest residues which can be targeted (especially non-Cys residues) maybe of assistance to others.

3) The paper will immensely benefit from discussing weaknesses. a): The complete modification of KRASG13C happens in high pH (9.5!!) which is of a significant concern. Do the authors envision a solution to the pH that is required for the covalent modification to take place. b): Do the authors see hope of designing something that can cross the membranes. c): These nucleotides will have the possibility to bind to other small GTPases or enzymes how is that going to affect this strategy. In particular: Other Ras superfamily member GTPases naturally have residue C13 equivalent 13 in Ras, such as Arl4a, RheBL1, Rab21 mentioned here. Have the authors tested the reactivity of edaGDP or analogs in these recombinant proteins? Is it possible that these compounds can be converted by NKDs to the GTP form?

4) There is no data to demonstrate that KRASG13C-effector binding is directly disrupted by eda GDP or others in vitro or in cells. Can addition of edGDP disrupt KRASG13C-effector interaction esp at lower pH?

5) Both structures lacked Mg^2+^ and is this the reason for the downstream effects?

6) It is important for the authors to convincingly demonstrate that their approach is better than the current state of the art esp. when there are options like SOS inhibitors and pan RAS inhibitors with good oral bioavailability which didn't exhibit overt toxicity and present a exploitable therapeutic window.

7) Related: I don't see an SOS mediated exchange using those compounds. The authors only show that SOS does not work on modified KRas. I find it difficult to deduce the time and the efficiency of covalently binding to KRasg13C in the presence of other nucleotides and in the presence of SOS. What kind of concentrations and how long will it take if you have in one cuvette physiological amounts of GTP/GDP and your compounds and Ras and SOS. This experiment would improve the manuscript significantly.

8) Perhaps the authors can already test if their derivatives would still work on patients who has gained resistance mutations like Y96C, H95Dor R68S with KRASG13C.

9) Please include the total protein blots for all the phospho-proteins tested in all the westerns. pS338 is not a reliable marker for checking the activation of CRAF. There are instances where CRAF is 338 phosphorylated but not activated. Ideally the authors should look at MEK phosphorylation. I appreciate the effects at pERK1/2.

10) One should also be careful with the general cytotoxicity independent of KRASG13C as shown with SML-8-3-1 especially when they have to be employed at higher concentrations for reasonable modification of KRASG13C. Please note that this allele can be amplified in patients.

11) The authors need to show target engagement in cellular systems directly. As of now, at least one could try to do electroporation into KRASG13C expressing cells and check if the modified protein is dimerizing with the endogenous version and has implications in the downstream signaling.

*Reviewer #1 (Recommendations for the authors):*

Ras is the first discovered oncogene and KRAS is most frequently mutated isoform. Recent studies led to the development of mutation specific inhibitors, especially against the KRASG12C mutant. However, unfortunately the patients treated with Adagrasib or similar ones develop resistance due to further gain of function mutations and amplification of KRASG12C allele apart from mutations in the downstream signaling components. One of the oldest approaches is to compete with the nucleotide binding of RAS and it has for a long time remained difficult owing to the picomolar affinity of RAS for GTD/GDP. Gray and colleagues tried to overcome these issues by employing GDP derivatives that can undergo covalent reaction with disease specific mutations but the authors reported in their previous work that the issue with these derivatives was with the loss of reversible affinities for β modified derivatives for RAS of at least by 10000 fold compared GDP and GTP. Here the authors present novel GDP derivatives different from Gray and colleagues and demonstrate that they could lock KRASG13C, another important mutant of KRAS in GDP bound form with a multiple set of biochemical, structural and cellular assays. While I find the study scientifically interesting I have several concerns that diminish my enthusiasm for publication in *eLife*. As such the manuscript is more appropriate for a journal with focus on structural /Chemical biology.

1) Can the edaGDP derivative competes with 1 mM of GDP and GTP in covalently modifying KRAS G13C and the wild type KRAS (well to a lesser extent)? I see the authors also employed Cys light version of RAS like the Shokat group in some of their assays and perhaps wild type protein is probably better here as well as anyway they are performing MS analysis.

2) The complete modification of KRASG13C happens in high pH (9.5!!) which is of significant concern.

3) Gray and colleagues performed the "caging " approach to allow their derivatives for passive cellular uptake. They did show that these modified GDP derivatives inhibit atleast signalling and binding to RAS effectors in cells. As of now the authors electroporate KRAS protein into HeLa cells to evaluate their effects. They claim to be working on ways to protect esterification for cellular entry. We need such development and proof of principle experiments that this can at least be a "prodrug" that can be further developed. I would like to see assays performed in KRASG13C allele addicted cell lines. First, I'm also surprised that the WT-KRAS didn't activate MAPK signaling as WT protein should be GTP bound and thus have basal activity while purified or when introduced to cells, especially in excess.

4) There is no data to demonstrate that KRASG13C-effector binding is directly disrupted by eda GDP or others in vitro or in cells. Can addition of edGDP disrupt KRASG13C-effector interaction esp at lower pH?

5) Both structures lacked Mg^2+^ and is this the reason for the downstream effects?

6) It is important for the authors to convincingly demonstrate that their approach is better than the current state of the art esp. when there are options like SOS inhibitors and pan RAS inhibitors with good oral bioavailability which didn't exhibit overt toxicity and present a exploitable therapeutic window.

7) Perhaps the authors can already test if their derivatives would still work on patients who has gained resistance mutations like Y96C, H95Dor R68S with KRASG13C.

8) Please include the total protein blots for all the phospho-proteins tested in all the westerns. pS338 is not a reliable marker for checking the activation of CRAF. There are instances where CRAF is 338 phosphorylated but not activated. Ideally the authors should look at MEK phosphorylation. I appreciate the effects at pERK1/2.

9) One should also be careful with the general cytotoxicity independent of KRASG13C as shown with SML-8-3-1 especially when they have to be employed at higher concentrations for reasonable modification of KRASG13C. Please note that this allele can be amplified in patients.

10) The authors need to show target engagement in cellular systems directly. As of now, at least one could try to do electroporation into KRASG13C expressing cells and check if the modified protein is dimerizing with the endogenous version and has implications in the downstream signaling.

*Reviewer #2 (Recommendations for the authors):*

Suggestions and questions for the authors:

1. K-Ras wildtype was labeled by eda/pda/bdaGDP to a much lower extent (Figure 1D, E). however, K-Ras(G12C) Cyslight was not labeled by these GDP analogs at all. Do the authors have any MS2 data to identify the site of modification in the case of K-Ras WT? There is growing interest in covalent adducts to RAS, so identifying the nearest residues which can be targeted (especially non-Cys residues) maybe of assistance to others.

2. Other Ras superfamily member GTPases naturally have residue C13 equivalent 13 in Ras, such as Arl4a, RheBL1, Rab21 mentioned here. Have the authors tested the reactivity of edaGDP or analogs in these recombinant proteins?

*Reviewer #3 (Recommendations for the authors):*

1 – The introduction would be improved if the authors mention the prevalence of G13C mutations. How much does it represent of all mutant Ras and if it is found in a specific cancer.

2 – The paper will immensely benefit from discussing the weakness …do the authors envision a solution to the pH that is required for the covalent modification to take place. Do the authors see hope of designing something that can cross the membranes. These nucleotides will have the possibility to bind to other small GTPases or enzymes how is that going to affect this strategy. is it possible that these compounds can be converted by NKDs to the GTP form?

3 – Experimentally: I don't see an SOS mediated exchange using those compounds. The authors only show that SOS does not work on modified KRas. I find it difficult to deduce the time and the efficiency of covalently binding to KRasg13C in the presence of other nucleotides and in the presence of SOS. What kind of concentrations and how long will it take if you have in one cuvette physiological amounts of GTP/GDP and your compounds and Ras and SOS. This experiment would improve the manuscript significantly.

---

## [Author Response]

Essential revisions:1) Can the edaGDP derivative competes with 1 mM of GDP and GTP in covalently modifying KRAS G13C and the wild type KRAS (well to a lesser extent)?I see the authors also employed Cys light version of RAS like the Shokat group in some of their assays and perhaps wild type protein is probably better here as well as anyway they are performing MS analysis.

Since we see only very minor modification of KRasWT even after 24 h at pH 9.5 (please see our answer and discussion to question 3 below), we have performed the experiments for KRasG13C (Cys light) below with equimolar concentrations of the competing nucleotides as well as 10x and 100x excess. As expected, the presence of other competing nucleotides slows down the reaction, but modification nevertheless still occurs. In fact, the object and reasoning of modifying the nucleotides at the 2’,3’-positions was to obtain nucleotides with comparable reversible affinities compared to the natural analogues as discussed in the manuscript. However, as stated throughout the manuscript, further improvement of the reactivity (k_inact_) is necessary to finally achieve reaction rates that are compatible with in vivo use.

We would also like to stress the point that the Cys-light version of KRas is the standard model protein used to develop covalent inhibitors and was used by other labs as well (as already stated by the reviewer) *^1^*.

2) Related: K-Ras wildtype was labeled by eda/pda/bdaGDP to a much lower extent (Figure 1D, E). however, K-Ras(G12C) Cyslight was not labeled by these GDP analogs at all. Do the authors have any MS2 data to identify the site of modification in the case of K-Ras WT? There is growing interest in covalent adducts to RAS, so identifying the nearest residues which can be targeted (especially non-Cys residues) maybe of assistance to others.

This is an excellent point and we have extensively tried to find the additional sites of modification in K‑RasWT using both the full-length protein as well as a truncated protein without the C-terminus (residues 1-169). Unfortunately, we have not been able to identify additional sites, which – considering the very low extent of modification and the difficulties we had previously in detecting the site of modification in KRasG13C by MS/MS experiments (Figure 3 —figure supplement 5) – is not unexpected.

3) The paper will immensely benefit from discussing weaknesses. a): The complete modification of KRASG13C happens in high pH (9.5!!) which is of a significant concern. Do the authors envision a solution to the pH that is required for the covalent modification to take place. b): Do the authors see hope of designing something that can cross the membranes. c): These nucleotides will have the possibility to bind to other small GTPases or enzymes how is that going to affect this strategy. In particular: Other Ras superfamily member GTPases naturally have residue C13 equivalent 13 in Ras, such as Arl4a, RheBL1, Rab21 mentioned here. Have the authors tested the reactivity of edaGDP or analogs in these recombinant proteins? Is it possible that these compounds can be converted by NKDs to the GTP form?

We have expanded the discussion our approach and we have modified the last paragraph of the conclusions accordingly. The paragraph now reads:

“Further optimization of the nucleotides to overcome the limitations regarding reactivity at physiological pH, as well as cell-permeability, are necessary and currently ongoing. In this respect, we are focusing on various linker designs based on the determined structure of the adduct, including cyclic linkers, to fine-tune the warhead's orientation and reactivity towards the cysteine at position 13. Additionally, protective esterification of the diphosphate moiety is currently being investigated to deliver prodrugs of the nucleotides into cells, which subsequently become activated by unspecific esterases.^32,33^ We are also investigating whether appropriate modification of the linker length and the electrophilic warhead would also make KRasG12C or other relevant mutants (e.g., HRasG12S in Costello syndrome)^34^ a possible target in a similar approach.”

Additional, we would like to stress:

a) Protein modification does NOT ONLY happen at pH 9.5, but as we clearly show it does also happen at pH 7.5 (~14% after 24h), pH 8 (~37% after 24h), pH 8.5 (~57% after 24h) and pH 9 (~90% after 24h) in the case of our best compound edaGDP, whereas KRasWT is only modified to a very small extend at pH > 9 (~3% at pH 9, ~8% at pH 9.5 after 24h), thus showing a rather good overall selectivity (Figure 1—figure supplement 5). For preparative labeling of KRasG13C (for subsequent crystallization, electroporation etc.), we chose pH 9.5 to ensure full modification as quickly as possible, and it will be undoubtedly necessary to increase the reactivity at physiological pH, however we are confident that we are on a very good track as pointed out above. In order to clarify this point, we have changed the manuscript accordingly (page 5-6).

b) Optimization of the nucleotide analogs with regard to cell permeability by synthesizing the corresponding prodrug derivatives, in which the negative charge of the phosphates is masked by esterification and which have already been described*^2-4^*, is currently being investigated.

c) Regarding the specificity: We cannot say anything for certain at this point and until we have cell-penetrable compounds. However, the problem regarding specificity is a well-known issue for covalently binding inhibitors also e.g. in the kinase field and this is also known and documented for the marketed drug osimertinib*^5^*.

Additionally, we have tested the reactivity of the nucleotides towards Rab21 (the only protein that is currently available to us with a cysteine at a position analogous to that in KRasG13C) and we see modification comparable to KRasG13C as expected. However, we currently cannot say what effect this might have in the context of inhibiting growth of KRasG13C dependent cancer cells or if this might have a negative (or potentially even positive) effect on this treatment option. Interestingly, knock-down of Rab21 has been shown to have positive effects in the context of human glioma cells *^6^*.

With respect to the last question (i.e. whether NKDs could potentially phosphorylate the diphosphate to the triphosphate form), we cannot say anything for certain at this point. However, even if this was the case and the nucleotides were phosphorylated within the cells, the intrinsic (and possibly GAP stimulated) hydrolysis will ultimately lead to the diphosphate-form.

4) There is no data to demonstrate that KRASG13C-effector binding is directly disrupted by eda GDP or others in vitro or in cells. Can addition of edGDP disrupt KRASG13C-effector interaction esp at lower pH?

We thank the reviewer for this suggestion and have now included direct evidence that the effector interaction is not possible for Ras-edaGDP. For this, we have performed pulldown experiments with GST-tagged RafRBD. The corresponding experiments have now been included in the manuscript (page 9, lines 206-213).

5) Both structures lacked Mg^2+^ and is this the reason for the downstream effects?

As we have stated in the manuscript, the lack of Mg^2+^-ions is, in all likelihood, a result of the crystallization conditions containing fluoride in both cases. This presumably leads to precipitation of poorly soluble MgF_2_. Additionally, we systematically and thoroughly measured the nucleotide affinities for the modified nucleotides in comparison to GDP/GTP, and these clearly show that affinities are essentially unchanged in case of the modified nucleotides, which is incompatible with significantly altered Mg^2+^-affinities. Additionally, we also measured the rate of intrinsic GTP-hydrolysis for KRasG13C-edaGTP, which is similar to the wild-type protein and also incompatible with a significantly altered Mg^2+^-affinity.

Nonetheless, we have also measured the Mg^2+^-affinity (Author response image 1), and we could not observe a difference between KRasWT and KRasG13C. These measurements were carried out as described previously*^7^*. Briefly, 1 μM KRas:mantdGDP or KRasG13C:mantdGDP were titrated against increasing concentrations of MgCl2 (1, 2.5, 5, 7.5, 10, 15, 20, and 25 μM) and the resulting curves were analysed by plotting the relative fluorescence intensity versus MgCl2 concentration.

**Author response image 1. sa2fig1:** Mg^2+^ affinity assay. (A, B) Fluorescence titration of KRasWT:mantdGDP or KRasG13C:mantdGDP with increasing concentrations of Mg^2+^ (1, 2.5, 5, 7.5, 10, 15, 20, and 25 μM); (C) The resulting curves were analysed by plotting the relative fluorescence intensity vs. MgCl2 concentration.

6) It is important for the authors to convincingly demonstrate that their approach is better than the current state of the art esp. when there are options like SOS inhibitors and pan RAS inhibitors with good oral bioavailability which didn't exhibit overt toxicity and present a exploitable therapeutic window.

We respectfully disagree that options are available, like SOS inhibitors and panRas inhibitors, since none of these has proven effective in clinical settings. It is correct that there are several studies as indicated by the reviewers (e.g. BI-1701963 and others), but at least until one approach has proven to be effective in patients, we feel that additional ideas and approaches are urgently needed (including our approach on using these covalently binding nucleotides). It is highly likely that even if an effective compound were available for treatment, upcoming resistance mutations would necessitate additional means of targeting Ras.

Additionally and importantly, the approaches mentioned above are not specific and do not allow direct targeting of a specific mutant of Ras (e.g. KRas G13C, which has NOT been addressed at all until now). We, therefore, feel that also under consideration of the current knowledge and available inhibitors, our study is an important additional contribution to the field, just like the studies on pan-Ras inhibitors, SOS-inhibitors, RasG12C inhibitors or PDEdelta inhibitors.

7) Related: I don't see an SOS mediated exchange using those compounds. The authors only show that SOS does not work on modified KRas. I find it difficult to deduce the time and the efficiency of covalently binding to KRasg13C in the presence of other nucleotides and in the presence of SOS. What kind of concentrations and how long will it take if you have in one cuvette physiological amounts of GTP/GDP and your compounds and Ras and SOS. This experiment would improve the manuscript significantly.

See point 1, corresponding experiments have been added to the manuscript.

8) Perhaps the authors can already test if their derivatives would still work on patients who has gained resistance mutations like Y96C, H95Dor R68S with KRASG13C.

It would indeed be interesting to test the effect of additional mutations, however, this is beyond the scope of this work.

Additionally, the reported mutations at position 96, 95 and 68 are all located within the switch-II pocket (and not the nucleotide-binding region) and are reported as a result of treatment with switch-II-pocket inhibitors that target the RasG12C mutation*^8, 9^*. Thus, it is highly likely that these will not have an effect on nucleotide binding, and most likely other resistance mutations will be of relevance upon using nucleotide-competitive inhibitors described in this publication. Possibly, the high conservation of the nucleotide-binding pocket could eventually even be an advantage regarding emerging resistances because only very few mutations might be generally tolerated, but this hypothesis will have to be tested and challenged in the future.

9) Please include the total protein blots for all the phospho-proteins tested in all the westerns. pS338 is not a reliable marker for checking the activation of CRAF. There are instances where CRAF is 338 phosphorylated but not activated. Ideally the authors should look at MEK phosphorylation. I appreciate the effects at pERK1/2.

The total protein blots for all the phospho-proteins tested in the western blots were already present in the original submission (Goebel et al_Source_data_file_WB.xlsx). Please let us know if anything else is missing.

We are aware that there are findings that phosphorylation at Ser-338 can be independent of c-Raf activity*^10^*. However, our Western blots clearly and unambiguously indicate a correlation between phospho-cRaf, phospho-pErk and the KRas concentration, i.e. a correlation between Ras, Raf and Erk activity (which is undoubtedly a good marker). Thus, we see no point in doing these additional experiments for MEK or c-Raf (Tyr340/341) activation that would require disproportionate additional effort and time for something that has already been clearly answered.

10) One should also be careful with the general cytotoxicity independent of KRASG13C as shown with SML-8-3-1 especially when they have to be employed at higher concentrations for reasonable modification of KRASG13C. Please note that this allele can be amplified in patients.

We agree that toxicity will need to be evaluated at a later point in time. Actually, in the design of the nucleotides, we purposely did not use chloroacetamides (as in SML-8-73-1), but rather less reactive and biocompatible acrylamides, since these are well known and have been shown to be tolerated in many approved drugs(e.g. osimertinib, afatinib to name just two clinically relevant examples of covalent drugs).

11) The authors need to show target engagement in cellular systems directly. As of now, at least one could try to do electroporation into KRASG13C expressing cells and check if the modified protein is dimerizing with the endogenous version and has implications in the downstream signaling.

The reviewer brings up an excellent point and we have indeed tried to do the experiment as suggested and probed extensively for target engagement in a cellular context by electroporating the modified nucleotides into cells. However, presumably because of the relatively low reactivity at physiological pH and undefined (probably low) concentrations of the nucleotides in cells upon electroporation, we could not observe an effect. Additionally, available KRas G13C cell lines such as H1355 (BRCA2: p.Gly2044Asp (homozygous), KRAS: p.Gly13Cys (heterozygous), TP53: p.Glu285Lys (homozygous)) and H1734 (ATM: p.Gln65Pro (heterozygous), KRAS: p.Gly13Cys (heterozygous), RB1: p.Ser127Ile (homozygous), TP53: p.Arg273Leu (homozygous)) are heterozygous for the G13C mutation and also generally contain additional mutations that complicate any conclusions drawn from the respective experiments (e.g. can any effect be expected at all against the background of the additional mutations and what percentage of modification of KRasG13C is required for this?). We are, therefore, currently working on establishing a homozygous Ba/F3 cell line that is dependent on the KRasG13C mutation that will be better suited for these experiments, but although we have spent some effort on this already, it has not been successful yet. For these reasons, we used the assay as described in the current manuscript and electroporated the already modified protein.

As stated above, we are also working on protected nucleotides by esterification of the phosphates that allow passage of the cell membrane. However, the synthesis of these nucleotides containing modifications at both the phosphates and the ribose is far beyond the scope of the current manuscript.

In addition to the comments we have addressed above, we would like to add a few more points:

Reviewer 1 asked why there is no upregulation observed in the cellular electroporation experiments for KRasWT, since it should be GTP-bound after purification (question #3). In our hands however, KRasWT is GDP-bound after purification as can be clearly seen in isocratic HPLC analyses and thus, no effect is to be expected. Even KRasG13C is GDP-bound after the purification (all purification buffers contain 10µM GDP), however the increased intrinsic nucleotide exchange rate (shown in Figure 3A) of this mutant even in the absence of any stimulus presumably leads to the activation observed in cells.

Reviewer 3 suggested to elaborate on the significance of the G13C mutation within the introduction. We have added a short paragraph to the introduction:

“Whereas position 12 is the most commonly mutated residue, G13 is the second most common mutation (14% of tumors harbor a mutation at this position) and in 6% of these cases, an acquired cysteine is found.*^11, 12^* In lung cancer, the prevalence of the G13C mutation is 3%, which is equivalent to approximately 7000 individuals in the US per year.*^11, 13”^*

References

[1] Ostrem, J. M., Peters, U., Sos, M. L., Wells, J. A., and Shokat, K. M. (2013) K-Ras(G12C) inhibitors allosterically control GTP affinity and effector interactions, Nature 503, 548-+.

[2] Mehellou, Y., Rattan, H. S., and Balzarini, J. (2018) The ProTide Prodrug Technology: From the Concept to the Clinic, Journal of medicinal chemistry 61, 2211-2226.

[3] Meier, C. (2017) Nucleoside diphosphate and triphosphate prodrugs – An unsolvable task?, Antivir Chem Chemother 25, 69-82.

[4] Meier, C., Meerbach, A., and Balzarini, J. (2004) Cyclosal-pronucleotides--development of first and second generation chemical trojan horses for antiviral chemotherapy, Front Biosci 9, 873-890.

[5] Finlay, M. R., Anderton, M., Ashton, S., Ballard, P., Bethel, P. A., Box, M. R., Bradbury, R. H., Brown, S. J., Butterworth, S., Campbell, A., Chorley, C., Colclough, N., Cross, D. A., Currie, G. S., Grist, M., Hassall, L., Hill, G. B., James, D., James, M., Kemmitt, P., Klinowska, T., Lamont, G., Lamont, S. G., Martin, N., McFarland, H. L., Mellor, M. J., Orme, J. P., Perkins, D., Perkins, P., Richmond, G., Smith, P., Ward, R. A., Waring, M. J., Whittaker, D., Wells, S., and Wrigley, G. L. (2014) Discovery of a potent and selective EGFR inhibitor (AZD9291) of both sensitizing and T790M resistance mutations that spares the wild type form of the receptor, Journal of medicinal chemistry 57, 8249-8267.

[6] Ge, J., Chen, Q., Liu, B., Wang, L., Zhang, S., and Ji, B. (2017) Knockdown of Rab21 inhibits proliferation and induces apoptosis in human glioma cells, Cell Mol Biol Lett 22, 30.

[7] John, J., Sohmen, R., Feuerstein, J., Linke, R., Wittinghofer, A., and Goody, R. S. (1990) Kinetics of interaction of nucleotides with nucleotide-free H-ras p21, Biochemistry 29, 6058-6065.

[8] Awad, M. M., Liu, S., Rybkin, II, Arbour, K. C., Dilly, J., Zhu, V. W., Johnson, M. L., Heist, R. S., Patil, T., Riely, G. J., Jacobson, J. O., Yang, X., Persky, N. S., Root, D. E., Lowder, K. E., Feng, H., Zhang, S. S., Haigis, K. M., Hung, Y. P., Sholl, L. M., Wolpin, B. M., Wiese, J., Christiansen, J., Lee, J., Schrock, A. B., Lim, L. P., Garg, K., Li, M., Engstrom, L. D., Waters, L., Lawson, J. D., Olson, P., Lito, P., Ou, S. I., Christensen, J. G., Janne, P. A., and Aguirre, A. J. (2021) Acquired Resistance to KRAS(G12C) Inhibition in Cancer, N Engl J Med 384, 2382-2393.

[9] Goebel, L., Müller, M. P., Goody, R. S., and Rauh, D. (2020) KRasG12C inhibitors in clinical trials: a short historical perspective, RSC Medicinal Chemistry.

[10] Oehrl, W., Rubio, I., and Wetzker, R. (2003) Serine 338 phosphorylation is dispensable for activation of c-Raf1, J Biol Chem 278, 17819-17826.

[11] Forbes, S. A., Beare, D., Gunasekaran, P., Leung, K., Bindal, N., Boutselakis, H., Ding, M., Bamford, S., Cole, C., Ward, S., Kok, C. Y., Jia, M., De, T., Teague, J. W., Stratton, M. R., McDermott, U., and Campbell, P. J. (2015) COSMIC: exploring the world's knowledge of somatic mutations in human cancer, Nucleic Acids Res 43, D805-811.

[12] Visscher, M., Arkin, M. R., and Dansen, T. B. (2016) Covalent targeting of acquired cysteines in cancer, Curr Opin Chem Biol 30, 61-67.

[13] Burge, R. A., and Hobbs, G. A. (2022) Not all RAS mutations are equal: A detailed review of the functional diversity of RAS hot spot mutations, Advances in cancer research 153, 29-61.